# Dickkopf1 fuels inflammatory cytokine responses

Nikolai P. Jaschke [1✉], Sophie Pählig[1], Anupam Sinha[1,2], Timon E. Adolph [3], Maria Ledesma Colunga[1], Maura Hofmann[1], Andrew Wang[4,5], Sylvia Thiele[1], Julian Schwärzler[3], Alexander Kleymann[6], Marc Gentzel [7], Herbert Tilg [3], Ben Wielockx [2], Lorenz C. Hofbauer[1], Martina Rauner [1], Andy Göbel[1] & Tilman D. Rachner[1]

Many human diseases, including cancer, share an inflammatory component but the molecular underpinnings remain incompletely understood. We report that physiological and pathological Dickkopf1 (DKK1) activity fuels inflammatory cytokine responses in cell models, mice and humans. DKK1 maintains the elevated inflammatory tone of cancer cells and is required for mounting cytokine responses following ligation of toll-like and cytokine receptors. DKK1-controlled inflammation derives from cell-autonomous mechanisms, which involve SOCS3-restricted, nuclear RelA (p65) activity. We translate these findings to humans by showing that genetic DKK1 variants are linked to elevated cytokine production across healthy populations. Finally, we find that genetic deletion of DKK1 but not pharmacological neutralization of soluble DKK1 ameliorates inflammation and disease trajectories in a mouse model of endotoxemia. Collectively, our study identifies a cell-autonomous function of DKK1 in the control of the inflammatory response, which is conserved between malignant and non-malignant cells. Additional studies are required to mechanistically dissect cellular DKK1 trafficking and signaling pathways.

[1] Department of Medicine III & Center for Healthy Aging, Technische Universität Dresden, Dresden, Germany. [2] Institute of Clinical Chemistry and Laboratory Medicine, Technische Universität Dresden, Dresden, Germany. [3] Department of Internal Medicine I, Gastroenterology, Hepatology, Endocrinology, and Metabolism, Innsbruck Medical University, Innsbruck, Austria. [4] Department of Medicine (Rheumatology, Allergy & Immunology), Yale University School of Medicine, New Haven, CT, USA. [5] Department of Immunobiology, Yale University School of Medicine, New Haven, CT, USA. [6] Division of Rheumatology, Department of Medicine III, Technische Universität Dresden, Dresden, Germany. [7] Molecular Analysis - Mass Spectrometry, Center for Molecular and Cellular Bioengineering (CMCB), Technische Universität Dresden, Dresden, Germany. ✉email: nikolai.jaschke@uniklinikum-dresden.de

Dickkopf1 (DKK1) was originally identified in *Xenopus laevis* as a regulator of embryonic head development[1]. Subsequent studies revealed additional functions of DKK1 such as control of bone metabolism, hematopoietic regeneration, epithelial renewal and wound healing[2–5]. Soluble DKK1 mediates most of its effects via canonical Wnt-signaling antagonism, although Wnt-independent DKK1 biology has been described as well[6,7]. Under homeostatic conditions, DKK1 is mainly expressed and secreted by osteoblasts and osteocytes, whereas elevated DKK1 expression is observed across multiple cell types and tissues in various diseases including malignancy[8–10]. Increased circulating DKK1 levels in cancer derive from both the tumor as well as the host and are frequently associated with a poor prognosis[8,11–13]. Blocking soluble, extracellular DKK1 confers heightened antitumor immunity in preclinical models[13–16]. These observations have paved the way for clinical trials investigating the utility of DKK1-neutralizing antibodies in the treatment of human cancer[16]. Although most diseases linked to DKK1 overproduction share an inflammatory component, the potential contributions of DKK1 to such changes remain incompletely understood. In this study, we identify a cell-autonomous function of DKK1 in the control of the inflammatory response, which is conserved between malignant and non-malignant cells.

## Results

### DKK1 expression associates with inflammation in human cancer

DKK1 is frequently overexpressed by malignant cells and inflammation is a hallmark of cancer[8,17,18]. Thus, we first screened publicly available RNA sequencing data from primary human tumor tissues (The Cancer Genome Atlas Project, TGCA)[19] for potential associations between transcript levels of DKK1 and different cytokines. We found that *DKK1* expression was positively correlated with inflammatory cytokines and chemokines such as interleukin 1 beta (*IL1B*), interleukin 6 (*IL6*), chemokine (C-X-C motif) ligand 8 (*CXCL8*, also referred to as IL8), interleukin 18 (*IL18*), and chemokine (C-C motif) ligand 2 (*CCL2*, also referred to as MCP1) across multiple tumor types (Fig. 1a and Fig. S1a). The strongest associations were observed in prostate cancer tissue (Fig. 1a). This finding was confirmed in an independent cohort of patients (Fig. 1b and S1 b). In contrast, correlations between DKK1 and cytokines mainly deriving from or acting on lymphocytes such as IL2, IL4, or IL17 were weaker or absent in tumor tissues (Fig. 1a and S1c). Screening of a panel of human epithelial cancer cell lines revealed that the highest *DKK1* and *IL1B* mRNA expression coincided in PC3 prostate cancer cells (Fig. 1c). We, therefore, focused on this cell line for further in vitro studies as a reductionist model for pathological DKK1 activity. Suppression of *DKK1* expression in PC3 cells using two independent small interfering RNAs (siRNAs) (Fig. 1d and Fig. S1d, herein further referred to as "siRNA#1" and "siRNA#2", respectively) followed by immunoblotting demonstrated DKK1-specific bands ranging from ~24–42 kDa (Fig. 1e), potentially reflective of different protein isoforms as previously reported[20]. DKK1 protein was most abundant in membrane and organelle-, but also readily detectable in nuclear fractions of wildtype, but not DKK1-deficient PC3 cells (Fig. 1f). Comparable results were obtained when DKK1 was ectopically overexpressed in T47D cells, which exhibit low basal DKK1 expression (Fig. S1e). Likewise, immunofluorescence staining followed by confocal microscopy located DKK1 abundance perinuclearly, likely corresponding to the endoplasmic reticulum, although the nuclear presence of DKK1 protein was again detected, specifically in wildtype cells (Fig. 1g). Taken together, these findings suggest that DKK1 expression correlates with an inflammatory phenotype in cancer.

### Suppression of DKK1 production curtails inflammatory cytokine expression in cancer cells

To better understand the biological consequences of pathological DKK1 activity, we performed bulk RNA sequencing from DKK1-competent and -deficient PC3 cells ($n = 3$/genotype), which revealed a total of 117 differentially regulated transcripts between genotypes (Fig. 2a). In DKK1-deficient cells, 32 transcripts were up-, and 85 were significantly downregulated compared to wild-type controls. These included *ALDH1A1*, a detoxifying enzyme, which has previously been shown to be regulated by DKK1[21] as well as the chemokine *CXCL8* (IL8). Using gene set enrichment analysis (GSEA), we assigned differentially regulated genes to biological programs (clusters) as defined by the KEGG database. This analysis showed that gene signatures related to inflammation, inflammatory cytokine-signalling and innate immunity were negatively enriched in the DKK1-deficient vs. wild-type dataset (Fig. 2b and S2a). To validate this finding, we profiled cytokine/chemokine transcripts using a PCR array, which demonstrated reduced mRNA expression of mediators such as *CXCL1, IL1B, CXCL8* (IL8) and *CSF1* (M-CSF) in DKK1-deficient PC3 cells (Fig. 2c). Individual quantification of selected cyto-/chemokines via qPCR and ELISA further confirmed these data (Fig. 2d, e) and we obtained comparable results in two independent tumor cell lines (DU-145 and MDA-MB-231) (Fig. 2f). Conversely, overexpression of DKK1 in cells with low basal DKK1 expression (T47D, LNCaP cells and primary murine bone marrow-derived macrophages; mBMDM) did not result in a consistent increase in cytokine transcription (Fig. S2b, c), suggesting that DKK1 is necessary, but not sufficient to drive inflammation.

### Cancer cells require DKK1 to mount inflammatory cytokine responses

Using different pro-inflammatory stimuli, we found that PC3 cells required DKK1 expression to effectively mount cytokine responses following ligation of toll-like and cytokine receptors (Fig. 3a–c and Fig. S3a, b). These effects were particularly prominent with bacterial lipopolysaccharides (LPS) as reflected by reduced IL1B and CXCL8 mRNA and protein expression in DKK1-deficient cells following exposure to the former (Fig. 3b-e). Comparable effects were observed in two independent cell lines (Fig. 3f). LPS-mediated induction of cytokine transcription in PC3 cells was TLR4- and Myd88-dependent (Fig. 3g), suggesting that pathogen-associated molecular pattern recognition mechanisms are comparable between tumor and immune cells.

Unexpectedly, supplementation of recombinant DKK1 failed to recover cytokine expression in DKK1-deficient cells (Fig. 3h). Likewise, antibody-mediated neutralization of soluble, extracellular DKK1 left cytokine transcription unchanged (Fig. 3i). These results were confirmed using recombinant DKK1 protein as well as a DKK1-neutralizing antibody from independent suppliers (Fig. 3j and Fig. S3c). The biological activity of the respective reagents was validated in primary murine osteoblasts, the principal recipients of endogenous DKK1 signaling. Here, recombinant DKK1 antagonized Wnt3a-mediated induction of Wnt-target genes such as lymphoid enhancer-binding factor 1 (*Lef1*) and osteoprotegerin (*Opg*), which was reversed by anti-DKK1 antibody treatment (Fig. 3k and Fig. S3d). Collectively, these data suggest that DKK1 promotes inflammatory cytokine expression in cancer cells in a cell-autonomous fashion.

### DKK1-controlled inflammation involves SOCS3-restricted nuclear RelA activity

DKK1 mediates most of its biological effects via competitive inhibition of LRP5/6 receptors, which antagonizes canonical Wnt-signaling[22]. Blockade of this pathway using XAV939 reduced Wnt-target gene expression, while

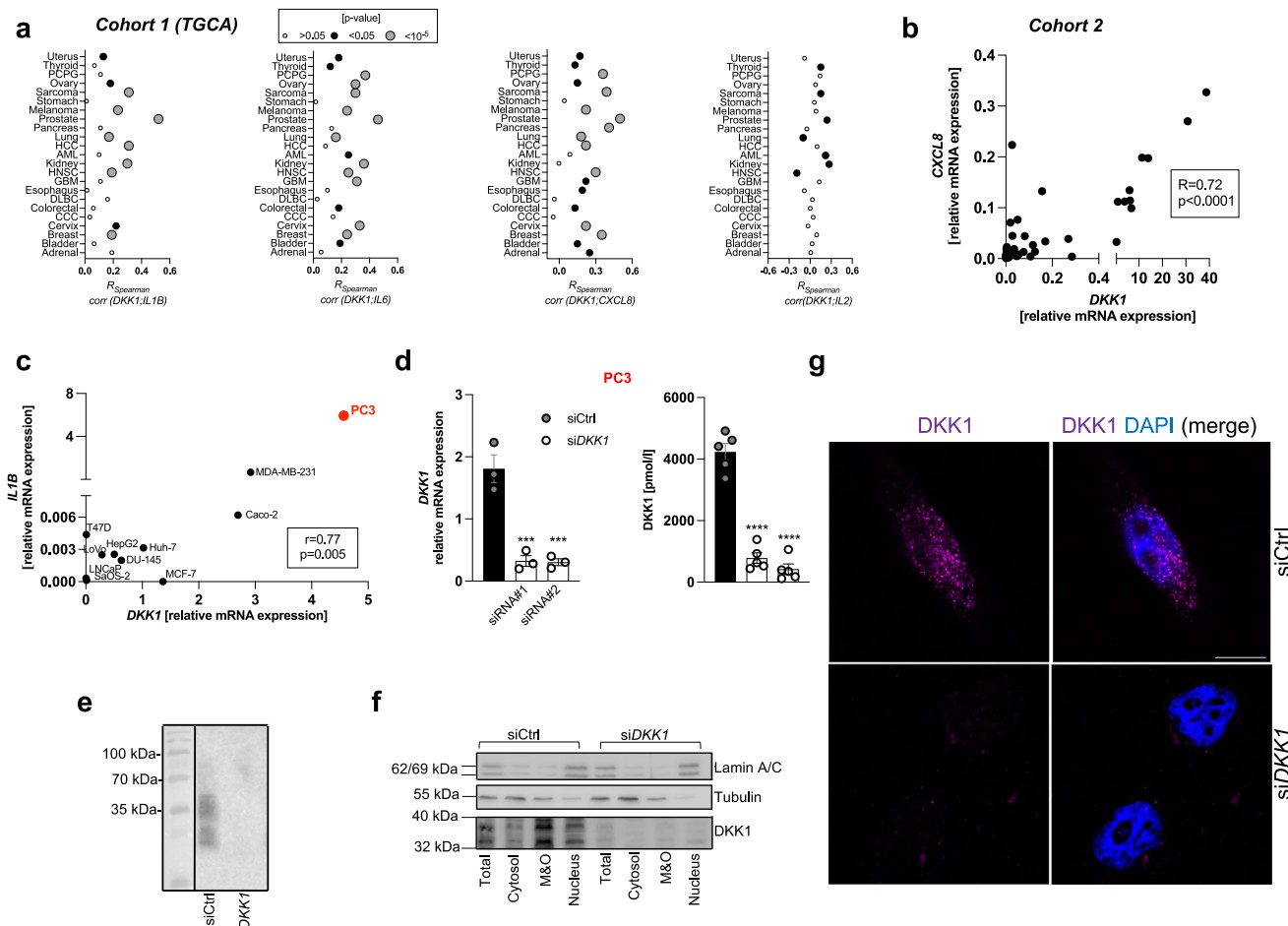

**Fig. 1 DKK1 expression associated with inflammation in human cancer. a** Correlations between transcript levels of *DKK1* and selected cyto-/chemokines in human tumor tissues visualized as bubble plots. Each dot indicates the Spearman correlation coefficient of the association between *DKK1* and cytokine/chemokine expression in the corresponding tissue (see X-axis). The size of the dot corresponds to the respective *p* value, whereby larger dots reflect smaller (significant) *p* values. No threshold for statistical significance was predefined. Data were extracted from the Cancer Genome Atlas Project (TGCA) dataset. The following abbreviations are shown: PCPG pheochromocytoma and paraganglioma, HCC hepatocellular carcinoma, AML acute myeloid leukemia, HNSC head and neck squamous cell carcinoma, GBM glioblastoma multiforme, DLBC diffuse large B-cell lymphoma, CCC cholangiocellular carcinoma. **b** Spearman correlation between *DKK1* and *CXCL8* (IL8) transcript levels in prostate cancer tissue from an independent cohort of patients. **c** Correlation between *DKK1* and *IL1B* mRNA expression in human epithelial cancer cell lines determined by qPCR using the ΔΔCT method. PC3 cells exhibiting the highest relative mRNA expression of both molecules are highlighted in red. **d** *DKK1* mRNA (left panel) and protein levels in cell culture supernatants (right panel) 48 h following transfection of PC3 cells with a non-targeting control oligonucleotide (siCtrl) or two independent siRNAs (siRNA#1 and siRNA#2) directed against *DKK1* mRNA (siDKK1) (*n* = 3–5/genotype). **e** Immunoblot of DKK1 from human cancer cell lysates. Specific bands were detected ranging from approximately 42–24 kDa. **f** DKK1 protein abundance in subcellular fractions from wildtype and DKK1-deficient cells determined by immunoblot. Representative blots are shown. **g** Immunofluorescence of DKK1 protein (visualized in magenta) in wildtype and DKK1-deficient PC3 cells. Nuclei are stained by DAPI. The scale bar (in white) shows 10 µM. *$p < 0.05$; **$p < 0.01$, ***$p < 0.001$, ****$p < 0.0001$. Data were expressed as mean ± SEM. [**a**, **b** Spearman correlation, **c** Pearson correlation, **d** one-way ANOVA with Holm–Sidak's post hoc test].

promoting *IL1B* transcription in wildtype but not DKK1-deficient PC3 cells (Fig. 4a and Fig. S4a). Likewise, the knockdown of LRP5, LRP6, or the combination of both did not affect cytokine expression in DKK1-competent cells (Fig. 4b and Fig. S4b). These results together with our previous observations implied that DKK1 promotes cytokine responses independent of Wnt-activity.

Unspecific manipulation of proteostasis using bortezomib (BZ), leptomycin B (LMB), and cycloheximide (CHX) suggested that DKK1-controlled inflammation may involve short-lived, anti-inflammatory proteins, which are degraded by the proteasome, exported from the nucleus and constantly replenished by ribosomal protein synthesis (Fig. 4c–e). Re-analysis of our RNAseq data using CARNIVAL[23] indicated that the corresponding molecule(s) may reside within the JAK/STAT- and/or NFkB-pathway (Fig. 4f). The JAK/STAT pathway harbors anti-inflammatory suppressor of

cytokine signaling (SOCS) proteins, of which SOCS1 and SOCS3 have been best studied in the context of inflammatory responses and TLR4 activation[24]. We first confirmed that SOCS1 and SOCS3 are short-lived proteins in PC3 cells (Fig. 4g and Fig. S4c). In all cellular compartments, SOCS3 was much more abundant than SOCS1 and the nuclear presence of the latter was only detectable following LMB treatment (Fig. 4h). In line with this observation, SOCS1 knockdown left cytokine expression unchanged, while suppression of SOCS3 yielded the recovery of cytokine transcription under both DKK1-deficient conditions (siRNA#1 and siRNA#2, respectively) (Fig. 4i and Fig. S4d–f). Likewise, SOCS3 knockdown restored LPS-triggered cytokine induction in DKK1-deficient cells, while evoking less prominent changes in DKK1-competent counterparts (Fig. 4j and Fig. S4g). These results were confirmed in an independent cell line (MDA-MB-231 (Fig. S4h).

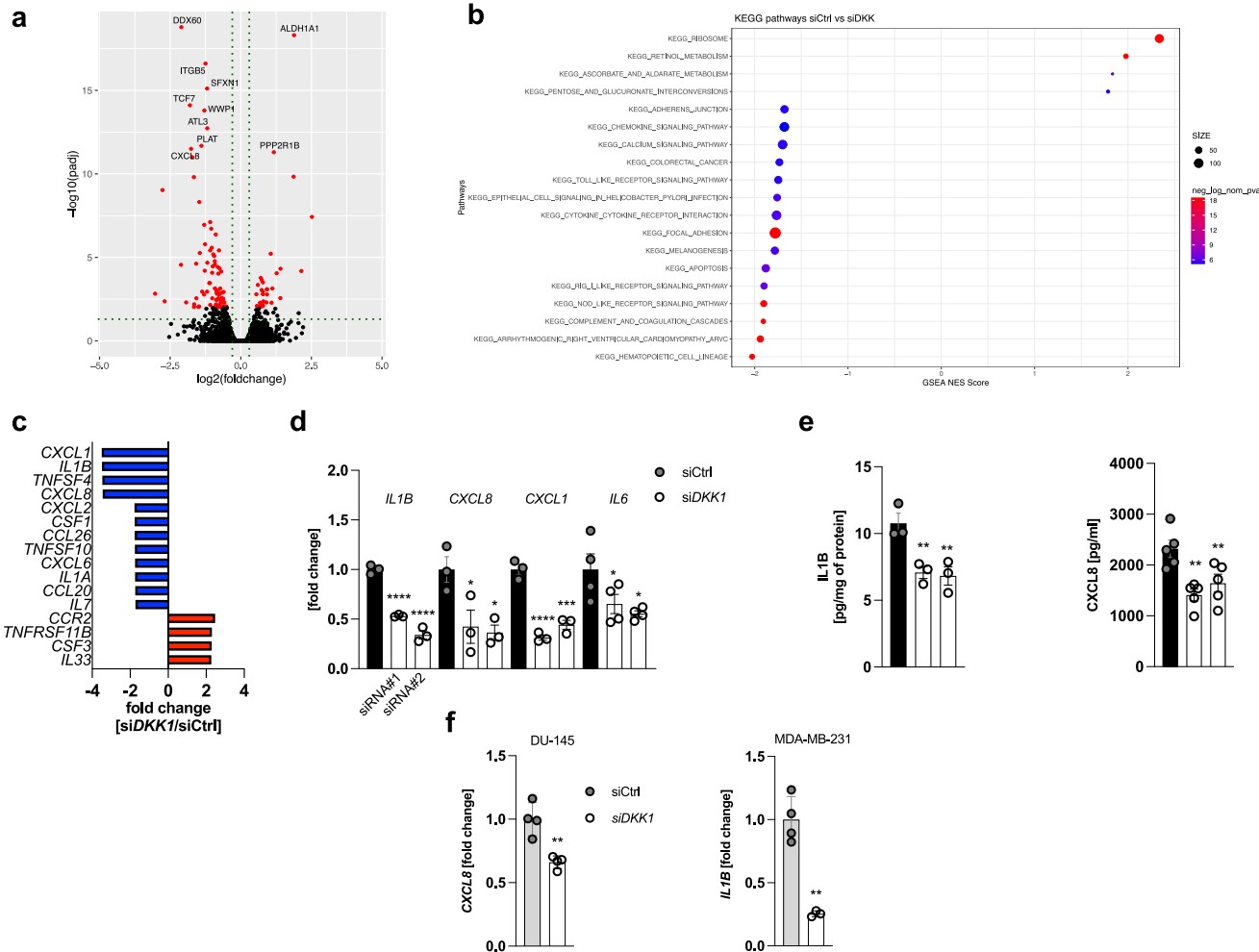

**Fig. 2 Suppression of DKK1 production curtails inflammatory cytokine expression in cancer cells. a** Differential expression analysis of RNAseq data from DKK1-deficient and wild-type PC3 cells (siDKK1 and siCtrl, respectively) visualized as a volcano plot. Significantly differentially expressed genes are highlighted in red ($n = 3$/genotype). **b** Results from Gene Set Enrichment Analysis (GSEA) visualized as a bubble plot. **c** Pro-inflammatory cytokine and chemokine PCR array from DKK1-deficient (siDKK1) and DKK1-competent (siCtrl) PC3 cells. The most differentially regulated transcripts are shown (mean of $n = 3$/genotype is shown). **d** Quantification of selected cyto-/chemokines by qPCR analysis in DKK1-deficient and -competent PC3 cells. Results for two independent DKK1-targeting siRNAs are shown (referred to as siDKK1#1 and siDKK1#2 and visualized as two separate bars across all figures) ($n = 3$/genotype). **e** IL1B and CXCL8 protein levels in total cell lysates and cell culture supernatants from PC3 cells with or without DKK1 knockdown. IL1B remained undetectable in cell culture supernatants ($n = 3$–5/genotype). **f** CXCL8 and IL1B mRNA expression in DU-145 prostate and MD-MB-231 breast cancer cells 48 h following siRNA-mediated DKK1 knockdown determined by qPCR. CXCL8 was chosen as a read-out in the former cells as IL1B expression was barely detectable ($n = 4$/genotype) *$p < 0.05$; **$p < 0.01$, ***$p < 0.001$, ****$p < 0.0001$. Data were expressed as mean ± SEM. [**d**, **e**: one-way ANOVA with Holm–Sidak's post hoc test, **f** two-tailed, unpaired student's $t$-test].

As most cytokines/chemokines regulated by DKK1 in cancer cells (including IL1B and CXCL8) are established NFkB-target genes and SOCS-molecules mediate ubiquitin-dependent degradation of various proteins including the canonical NFkB subunit RelA (=p65)[25–27], we next quantified protein levels of the latter by subcellular fractionation and immunoblotting. Nuclear RelA abundance was reduced in DKK1-deficient cells but increased following SOCS3 knockdown (Fig. 4k–m and Fig. S4i). Consistently, RelA overexpression conferred full restoration of cytokine mRNA levels (Fig. 4n and Fig. S4j). Neither suppression of inhibitor of kappa B alpha (IkBa), nor beta (IkBb) expression entailed similar alterations and phosphorylation of RelA itself was indistinguishable between genotypes (Fig. S4k–n), arguing that canonical mechanisms of RelA regulation did not contribute to the effects observed.

In summary, these data suggest that DKK1-controlled inflammation is Wnt-independent and involves nuclear RelA

activity, which is restricted by SOCS3. The precise mechanism of cell-autonomous DKK1 signaling remains to be defined.

**Genetic DKK1 variants are linked to boosted cytokine production in humans.** The regulation of inflammatory cytokine expression by high DKK1 expression in human cancer raised the question of whether physiological DKK1 activity confers similar functions. To explore this, we extracted single nucleotide polymorphism (SNP) and cytokine quantitative trait loci (cQTL) data common to DKK1 from the SNP database (dbSNP) and the 500FG cohort study (https://hfgp.bbmri.nl/), respectively[28,29]. As part of the Human Functional Genomics Project (HFGP), the 500FG study investigated genetic variants affecting cytokine production by combining genetic analysis and ex vivo challenges of leukocytes with infectious and non-infectious triggers[29]. Here, we found that DKK1 polymorphisms were associated with

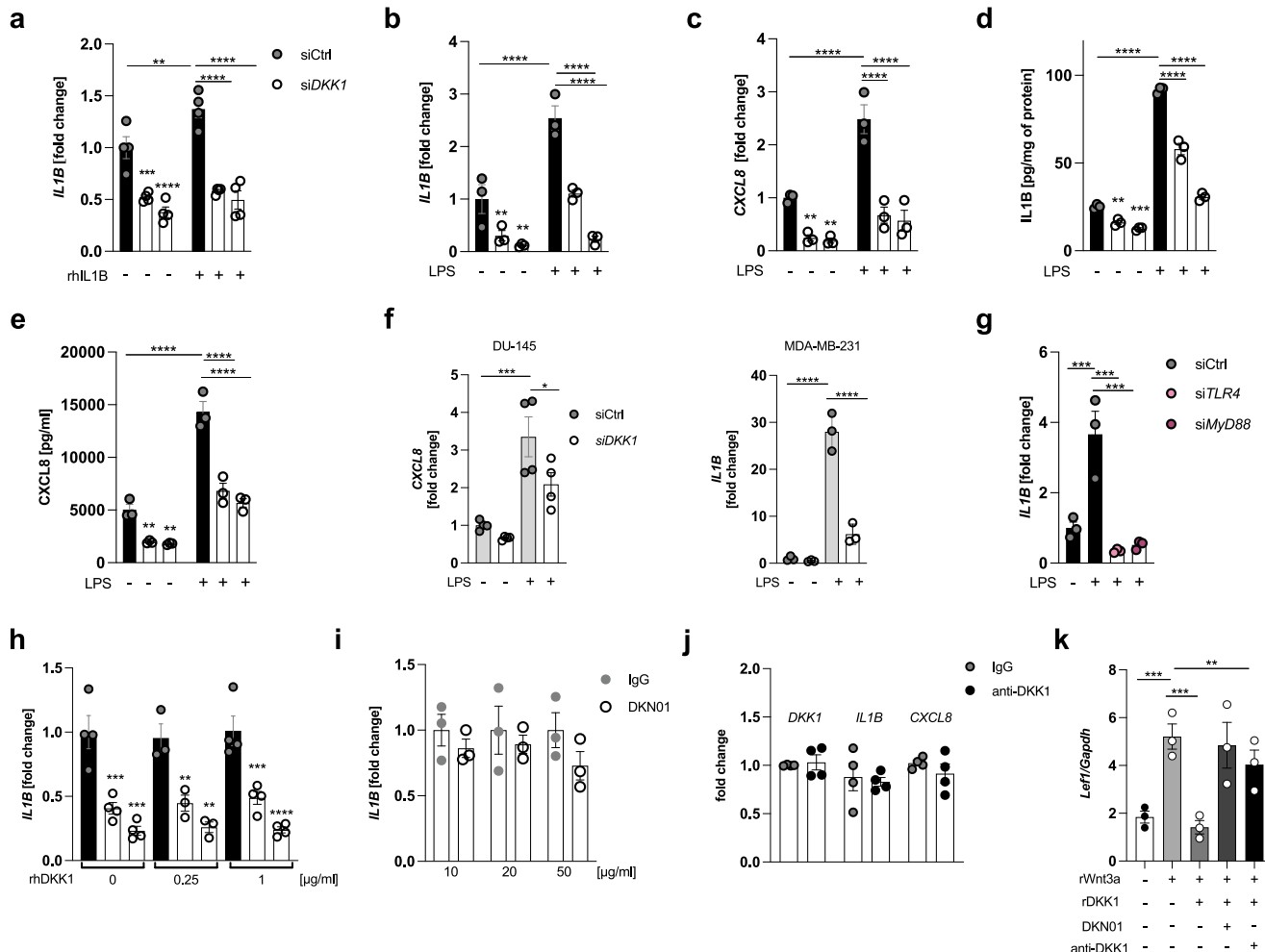

**Fig. 3 Cancer cells require DKK1 to mount inflammatory cytokine responses. a** *IL1B* transcript levels in DKK1-competent and -deficient PC3 cells following 24 h exposure to recombinant IL1B (rhIL1B) (20 ng/ml) or vehicle (PBS) (n = 4/group). **b, c** *IL1B* and *CXCL8* mRNA expression in DKK1-competent and -deficient PC3 cells exposed to LPS (1 μg/ml) or water for 24 h (n = 3/condition). **d, e** Corresponding protein levels of IL1B and CXCL8 in cell lysates and cell culture supernatants, respectively, determined by ELISA (n = 3/condition). **f** *CXCL8* and *IL1B* mRNA expression in DKK1-competent and -deficient DU-145 and MD-MB-231 cells treated with LPS (1 μg) for 24 h (n = 3–4/condition). **g** LPS-induced *IL1B* mRNA expression in wildtype or TLR4- and Myd88-deficient PC3 cells (n = 3). **h** *IL1B* transcript levels in DKK1-competent and -deficient PC3 cells treated with physiological (=amount of DKK1 secreted into cell culture supernatants) or excessive (four times higher than the former) concentrations of recombinant human DKK1 (rhDKK1) for 24 h (n = 3–4/condition). **i** *IL1B* mRNA expression in wild-type PC3 cells exposed to a DKK1-neutralizing antibody (DKN01) or isotype control (IgG4) at various concentrations for 24 h (n = 3). **j** *DKK1*, *IL1B* and *CXCL8* transcript levels in PC3 cells treated with an independent anti-DKK1 antibody (10 μg/ml) or isotype control for 24 h (n = 3). **k** mRNA levels of the Wnt-target gene *Lef1* in primary murine osteoblasts following treatment with recombinant Wnt3a (200 ng/ml) with or without recombinant DKK1 (250 ng/ml) and DKK1-neutralizing antibodies (DKN01 and anti-DKK1, respectively, both at 10 μg/ml) (n = 3/condition) *p < 0.05; **p < 0.01, ***p < 0.001, ****p < 0.0001. Data were expressed as mean ± SEM. [**a–k**: one-way ANOVA with Holm–Sidak's post hoc test).

increased secretion of pro-inflammatory cytokines such as Interferon-gamma (IFNγ), IL1B, and IL6 from peripheral blood mononuclear cells following exposure to *Borrelia, E. coli* and *Cryptococcus* or *Candida*, respectively (Fig. 5a and Table S1). Next, we used an independent dataset, which comprised pQTLs (protein Quantitative Trait Loci) associated with DKK1 from a study involving more than 10.000 individuals of European-descendent (https://www.omicscience.org/apps/pgwas/)[30]. We identified ten DKK1-associated variants linked to altered expression of various cytokines including *IL1B* (Fig. 5b). To study the effect size and direction (up- or downregulation) of these associations, we constructed a network using Cytoscape[31]. This network is bipartite, depicts pQTLs in green hexagons, and cytokines in ovals. Positive associations (DKK1 variants linked to increased cytokine expression) are visualized in red, negative associations (reduced cytokine expression) in blue and color

gradations reflect effects in between. Effect sizes are highlighted by the thickness of the respective arrows. Using this approach, we found that DKK1 variants were significantly associated with increased expression of pro-inflammatory cytokines and chemokines such as *IL1B, MCP1* or *CXCL11*, whereas other cytokines showed opposing trends (Fig. 5c and Supplementary Data 1). Of note, we identified three common DKK1 variants, which were shared between the two study populations (rs11001560, rs11815201 and rs1569198; highlighted in yellow), suggesting that they may confer a selective advantage.

**Circulating DKK1 levels are suppressed in response to infectious triggers.** Having established a genetic link between DKK1 and cytokine responses in healthy individuals, we next explored human DKK1 biology in the context of infectious disease. We

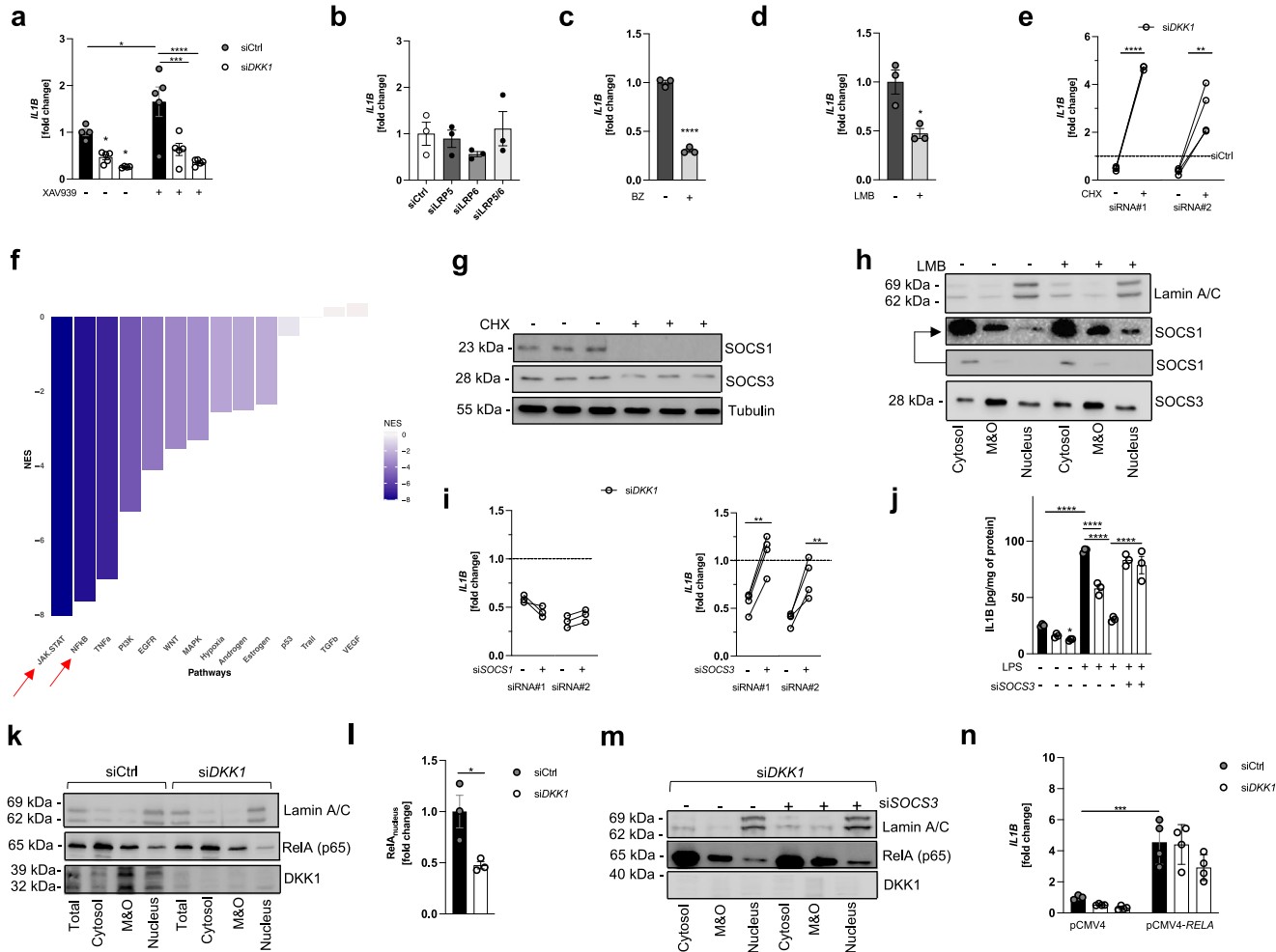

**Fig. 4 DKK1-controlled inflammation involves SOCS3-restricted nuclear RelA activity. a** *IL1B* mRNA levels in wildtype and DKK1-deficient PC3 cells following exposure to DMSO (−) or XAV939 (+) (5 μM) for 24 h (also see Fig. S3). The two DKK1-deficient conditions shown correspond to two independent siRNAs as in previous figures (n = 5/condition). **b** Transcript levels of *IL1B* in wild-type PC3 cells 48 h after siRNA-mediated knockdown of the two Wnt- and DKK1-receptors LRP5, LRP6, or the combination of both (also see Fig. S3) (n = 3). **c** *IL1B* mRNA expression in wild-type PC3 cells treated with the proteasome inhibitor bortezomib (BZ) at 100 nM for 6 h (n = 3). **d** *IL1B* transcript levels in wild-type PC3 cells treated with the CRM1 inhibitor leptomycin B (LMB) at 25 nM for 6 h (n = 3). **e** *IL1B* mRNA expression in DKK1-deficient PC3 cells exposed to the translation inhibitor cycloheximide (CHX) (1 μM) for 6 h. Both DKK1-targeting siRNAs (siRNA#1 and siRNA#2, respectively) are shown. The dashed line corresponds to *IL1B* expression of DMSO-treated PC3 cells transfected with non-targeting control oligonucleotides (siCtrl) (n = 3). **f** Normalized enrichment scores (NES) for signaling pathways linked to the pro-inflammatory response in DKK1-deficient vs. DKK1-competent PC3 cells. Results were generated using the RNAseq data from previous figures and CARNIVAL. **g** Representative SOCS1 and SOCS3 immunoblot from wild-type PC3 cells exposed to DMSO or Cycloheximide (CHX) for 6 h. **h** Representative immunoblot of subcellular fractions from wild-type PC3 cells treated with leptomycin B (LMB) or DMSO for 2 h. The upper SOCS1 lane is strongly contrast-stretched to aid visualization. **i** *IL1B* mRNA levels in DKK1-deficient cells exposed to siRNAs directed against SOCS1 (left panel) or SOCS3 (right panel). Results for two independent DKK1-targeting siRNAs are shown (siRNA#1 and siRNA#2, respectively) and normalized to siCtrl (n = 4/condition). **j** IL1B protein levels in total cells lysates of DKK1-competent and -deficient PC3 with or without SOCS3 knockdown 24 h following LPS exposure (1 μg/ml) (n = 3). **k, l** Representative immunoblot of subcellular fractions from wildtype (siCtrl) and DKK1-deficient (siDKK1) PC3 cells. Band intensities were quantified by ImageJ. **m** Representative immunoblot of subcellular fractions from DKK1-deficient cells transfected with (+) or without (−) siRNA directed against SOCS3 (siSOCS3). **n** *IL1B* mRNA levels in DKK1-competent and -deficient PC3 cells transfected with an empty backbone vector (pCMV4) or a plasmid encoding for human RelA (pCMV4-RELA) (n = 4/condition). *p < 0.05; **p < 0.01, ***p < 0.001, ****p < 0.0001. Data were expressed as mean ± SEM [**a, j, n**) one-way ANOVA with Holm–Sidak's post hoc test, **c–e, i** and **l** two-tailed, unpaired student's t-test].

recruited 27 hospitalized patients with acute pneumonia, of which the majority (91.7%) exhibited elevated (>0.06 mg/dl) procalcitonin levels (PCT), suggestive of an underlying bacterial infection (Fig. S5a)[32]. The median age of patients was 76 years (range: 30–88 years) and most were males (77.8%). Common comorbidities comprised arterial hypertension, coronary heart disease, chronic obstructive pulmonary disease and diabetes mellitus type II (44, 30, 37 and 41% of patients, respectively).

Circulating DKK1 levels during the acute, initial phase of pneumonia (day 1 or 2 of hospitalization) were characterized by considerable interindividual variability (range: 1.13–48.05 pmol/l) (Fig. S5b). DKK1 serum concentration increased modestly across the disease trajectory (+22%), which coincided with reductions in C-reactive protein levels (CRP), white blood cell count (WBC) body temperature (fever) as well as increases in circulating platelet numbers, markers of clinical recovery (Fig. S5c–g).

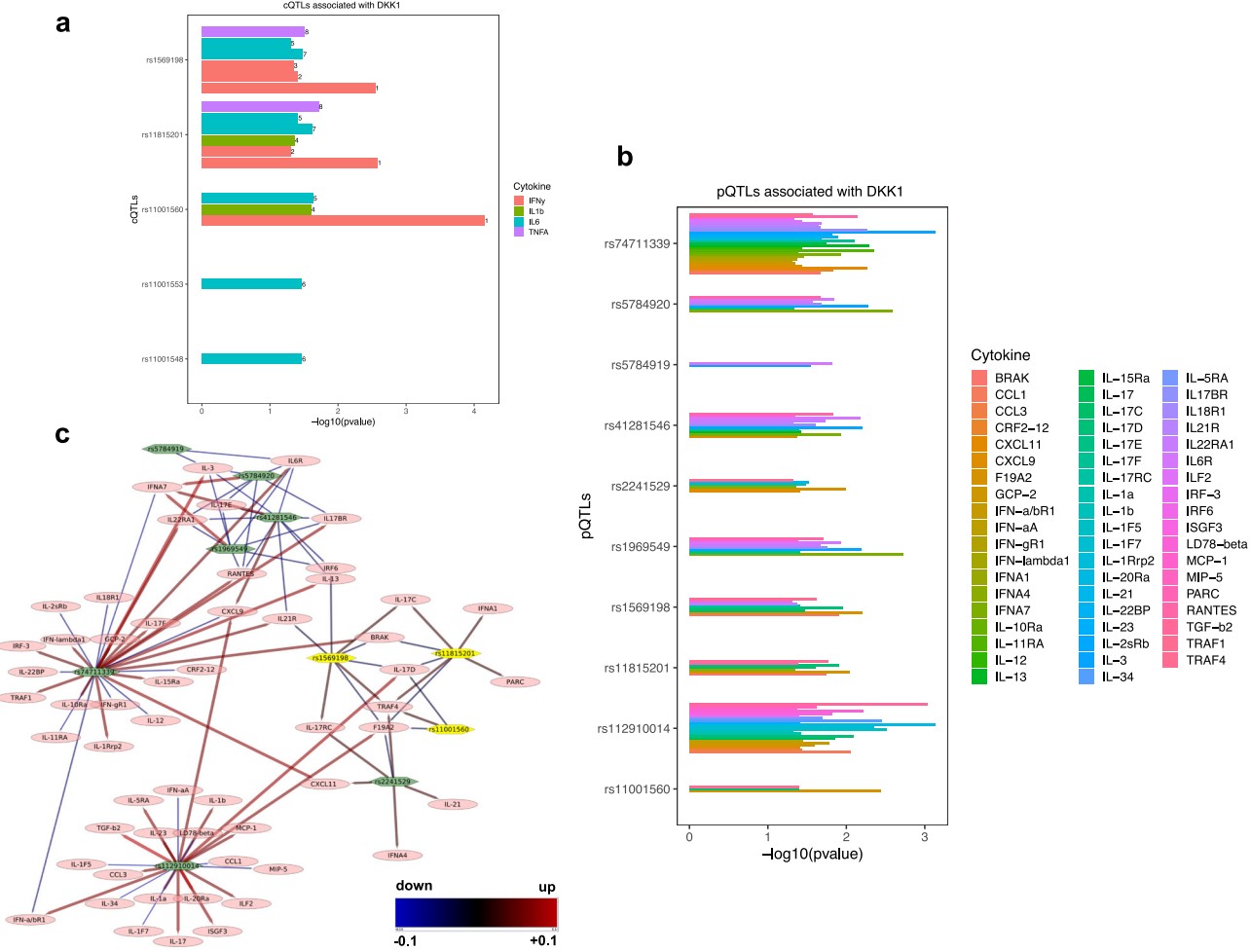

**Fig. 5 Genetic DKK1 variants are linked to boosted cytokine production in humans. a** Associations between genetic *DKK1* variants and cytokine expression in response to infectious triggers (cytokine quantitative trait loci, cQTLs) from the 500FG study. The numbers shown on the top of the bars correspond to distinct experimental setups used in the original study[29] (see Supplementary Data 1). **b** Associations between genetic DKK1 variants (protein quantitative trait loci; pQTLs) and the expression of various cytokines from an independent human cohort[30]. Log-transformed *p* values are shown. **c** A bipartite network depiction of the DKK1-associated pQTLs and various cytokines/chemokines. Green hexagonal nodes depict pQTLs, whereas cytokines are visualized as pink ovals. Positive associations (DKK1 variants linked to increased cytokine expression) are visualized in red and negative associations (reduced cytokine expression) in blue with color gradations in between. Effect sizes are highlighted by the thickness of the respective arrows. DKK1 variants shared between the two study populations are highlighted in yellow.

Circulating DKK1 levels at follow-up were correlated with platelet but not white blood cell counts (Fig. S5h, i), the former reflecting rich sources for DKK1[33–36].

In line with these observations, C57BL/6 J wild-type mice exhibited a pronounced drop in Dkk1 serum concentration at 3 h following a single injection of bacterial LPS (Fig. S6 a). Due to the rapid nature of these changes, we asked whether Dkk1 was being redistributed from the circulation in response to such perturbations. Quantification of Dkk1 protein in different tissues using an ELISA-based approach showed elevated Dkk1 levels in the spleen and the liver, but not in the bone or bone marrow in LPS-treated animals (Fig. S6b). Further assessment of serum kinetics revealed that Dkk1 concentration dropped within 1 h during the acute inflammatory response of the host and remained persistently suppressed, even after inflammation had resolved (Fig. S6c–e). Concurrently, Dkk1 protein abundance was elevated in the spleen and the bone marrow, which followed distinct kinetics and paralleled reductions in platelet counts, reminiscent of our findings in humans (Fig. S6f–h).

The mitogen-activated protein kinase (MAPK) p38 regulates Dkk1 in various experimental models[34,37–39]. This prompted us to

investigate if p38-signaling contributes to altered Dkk1 secretion in response to microbial inflammation. Phosphorylation (activation) of p38 in the spleen decreased in a time-dependent manner following LPS administration and pharmacological p38 inhibition amplified changes in circulating and splenic Dkk1 levels (Fig. S6i–k), suggesting that p38 activity serves as a rheostat for LPS-tuned Dkk1 secretion.

Taken together, these results suggest that suppression of circulating DKK1 levels is a shared feature of microbial inflammation in mice and humans.

**Genetic DKK1 deletion ameliorates inflammation and disease trajectories in a mouse model of endotoxemia.** Our results thus far suggested that inhibition of Dkk1 activity may be exploited to restrain overshooting cytokine production, a hallmark of sepsis. In contrast to the chronic inflammation observed in cancer, sepsis is characterized by an acute cytokine storm. Nonetheless, both entities share similar molecular effector mechanisms, including NFkB activation[18]. We thus asked whether the pro-inflammatory function of DKK1 is conserved between malignant and non-malignant cells, irrespective of the disease setting.

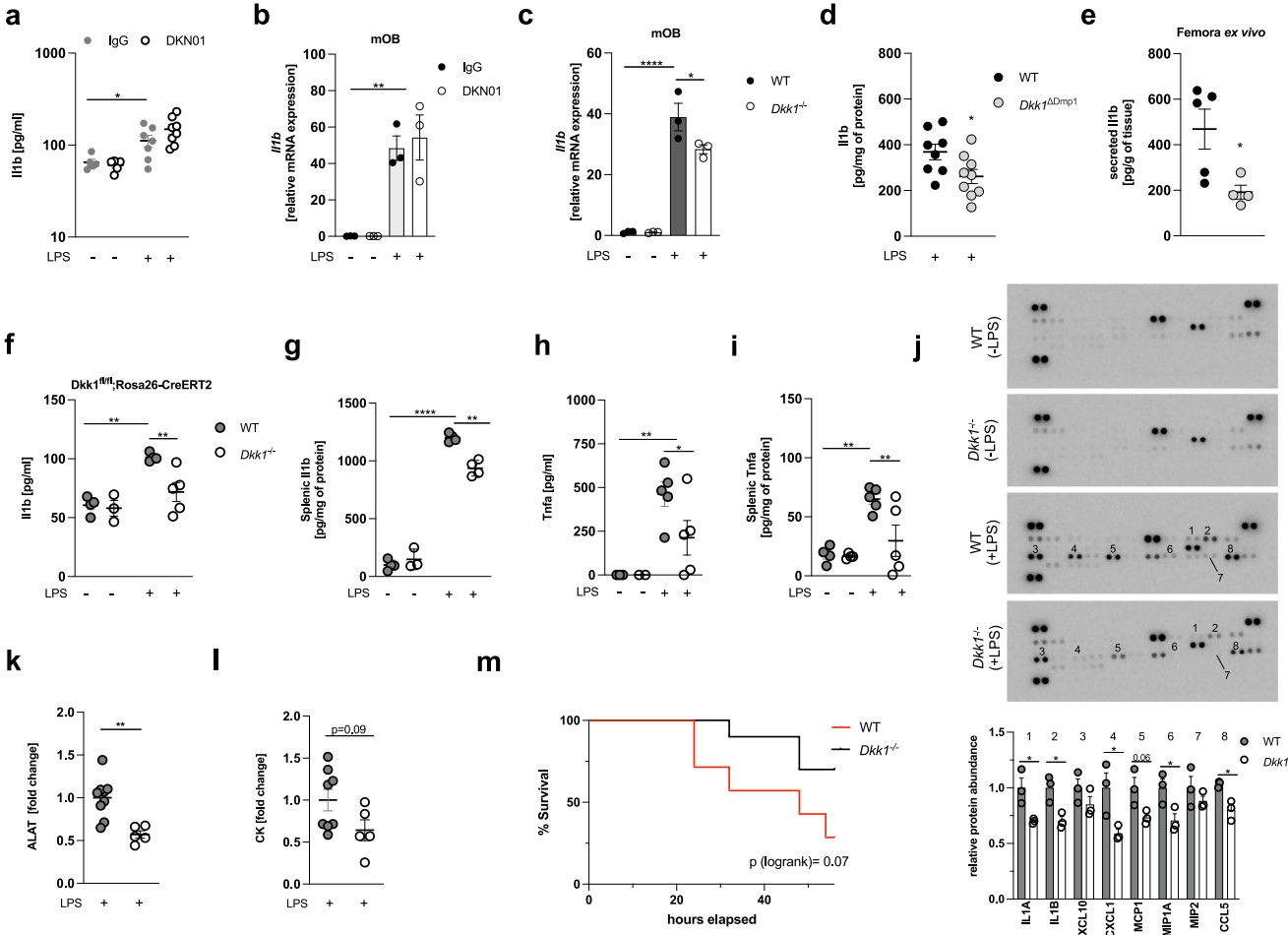

**Fig. 6 Genetic Dkk1 deletion ameliorates inflammation and disease trajectories in a mouse model of endotoxemia. a** Circulating Il1b levels in C57BL/6J mice 3 h following LPS (5 mg/kg) treatment. Eighteen hours prior to being challenged with LPS, mice received a single injection of a Dkk1-neutralizing antibody (DKN01, 5 mg/kg) or an equivalent dose of isotype control antibody (murine IgG) (n = 5–8/group). **b** Il1b mRNA expression in primary murine osteoblasts (mOB) differentiated from bone marrow precursors of C57BL/6J mice. Cells were differentiated for 7 days before being exposed to DKN01 (10 μg/ml) or isotype control (murine IgG; 10 μg/ml) for 18 h, followed by LPS treatment (1 μg/ml) for 6 h (n = 3/group). **c** Il1b mRNA expression in Dkk1 knock-out (isolated from bone marrow precursors of Dkk1−/− mice) or wild-type mOBs (obtained from littermate controls) treated with LPS (1 μg/ml) for 6 h (n = 3/group). **d** Il1b protein expression in bone lysates of mice with (Dkk1ΔDmp1) or without (littermate controls; referred to as wildtype or "WT") osteoblast/osteocyte-specific Dkk1 deletion treated with LPS (5 mg/kg) for 3 h. Bones were collected in PBS, flushed to remove the bone marrow, and total protein of the lysate was analyzed for Il1b concentration by ELISA (n = 8–9/group). **e** Il1b concentration in cell culture supernatants determined by ELISA following 24 h cultivation of the contralateral femur (without the bone marrow) from the same animals (n = 4–5/group). **f–i** Circulating and splenic tissue levels of Il1b and tumor necrosis factor-alpha (Tnfa) in Dkk1 knock-out mice (Dkk1−/−) or wild-type littermate controls (WT) 3 h following treatment with LPS or water (n = 3–5/group). **j** Cytokine proteome profiler array of spleen lysates. Membranes precoated with cytokine/chemokine-specific antibodies (in duplicates) were incubated with spleen lysates from Dkk1−/− mice or littermate controls. Differences between selected cyto-/chemokines were analyzed by quantification of dot intensities using ImageJ and expressed as fold-change (lower panel). Bar graphs show results from three biologically independent replicates (= 3 mice/genotype). **k, l** Circulating alanine aminotransferase (ALAT) and creatinine kinase (CK) levels in Dkk1−/− mice and littermate controls 30 h following LPS administration. Results are expressed as fold-change compared to LPS-treated wild-type controls (n = 5–8/group). **m** Survival of wildtype (n = 7) and Dkk1−/− mice (n = 10) treated with LPS (15 mg/kg). *p < 0.05; **p < 0.01; ***p < 0.001; ****p < 0.0001. Data were expressed as mean ± SEM. [a–c and f–l one-way ANOVA with Holm–Sidak's post hoc test, d, e and j–l) two-tailed, unpaired student's t-test].

Consistent with our previous observations, neutralization of soluble Dkk1 did not affect LPS-induced inflammation in vivo, irrespective of the dose of antibody applied (5 and 10 mg/kg as previously reported[14,15]) (Fig. 6a and Fig. S7a, b). Likewise, neither cells with high (murine osteoblasts; mOB), nor those with low (murine bone marrow-derived macrophages; mBMDM) basal Dkk1 expression exhibited changes in LPS-induced cytokine expression following exposure to anti-Dkk1 antibodies (Fig. 6b and Fig. S7c, d). In contrast, cytokine transcription in response to TLR4 ligation was reduced in Dkk1 knock-out (Dkk1−/−) osteoblasts as well as mBMDM isolated from mice with myeloid cell-specific Dkk1 deletion (referred to as Dkk1ΔLysM) (Fig. 6c and

S7f, g). Consistently, animals with osteoblast/osteocyte-specific Dkk1 deletion (Dkk1ΔDmp1) expressed less Il1b protein in bone tissue than wild-type littermate controls upon being challenged with LPS and femora of Dkk1ΔDmp1 mice maintained lower Il1b secretion when further cultivated ex vivo (Fig.6d, e). Likewise, timed global Dkk1 deletion (Dkk1fl/fl:Rosa26-CreERT2) yielding both local and systemic depletion of Dkk1 (hereafter referred to as Dkk1−/−) (Fig. S7h, i) ameliorated LPS-induced inflammation in the circulation and the spleen (Fig. 6f, g), the latter corresponding to the organ in which we had previously noted Dkk1 accumulation following TLR4 agonist exposure (see Fig. S6). These effects were not inflammasome-specific as Il1b and

tumor necrosis factor-alpha (Tnfa) levels were similarly decreased (Fig. 6h, i). Of note, unbiased profiling of spleen lysates revealed reduced expression of multiple cytokines and chemokines in $Dkk1^{-/-}$ animals (Fig. 6j). These changes followed a comparable pattern as those initially observed in our cancer cell model and were also reminiscent of correlations between DKK1 and cytokine transcripts in human tumor tissues. Specifically, LPS-treated $Dkk1^{-/-}$ mice exhibited decreased protein levels of Il1 alpha, Il1 beta, Cxcl1 (the murine homolog of CXCL8/IL8), Mcp1, Mip1 alpha, and Ccl5 compared to wild-type littermate controls (Fig. 6j), all of which share NFkB-response elements (https://www.bu.edu/nf-kb/gene-resources/target-genes/). Ameliorated inflammation upon genetic Dkk1 deletion was paralleled by signs of attenuated organ damage as evidenced by reduced alanine aminotransferase (ALAT) and creatinine kinase (CK) levels in the circulation (Fig. 6k, l). These changes conferred partial protection against LPS-induced mortality (Fig. 6m).

## Discussion

Inflammation is a high-cost/high-benefit trait: it aids in maintaining organismal homeostasis in response to environmental challenges such as invading pathogens while imposing considerable fitness costs entailed by unavoidable collateral tissue damage, high energetic demands and the risk of developing immunopathology[40,41]. Many of the beneficial but also detrimental effects of the inflammatory response are relayed by cytokines and chemokines, rendering the manipulation of these molecules an attractive therapeutic approach in the clinic. Our study demonstrates that both physiological (mice, primary murine cells, humans) and pathological (tumor cells) DKK1 activity fuels inflammatory cytokine production, which contributes to maladaptive inflammation in disease. This observation is consistent with other studies, which have reported a pro-inflammatory function of DKK1 in the context of allergic and non-allergic airway disease[34,35].

In our experimental models, soluble forms of DKK1 were dispensable for these functions. Previous studies have already pointed towards local DKK1 effects, which are independent of systemic DKK1 levels. Mice with osteoblast/osteocyte-specific DKK1 knock-out ($Dkk1^{\Delta Dmp1}$) exhibit diminished skeletal DKK1 expression, while circulating DKK1 levels are unchanged. Nonetheless, bone mass is strongly increased in these animals and the corresponding changes are comparable to those observed in $Dkk1^{\Delta Osx}$ mice, in which DKK1 is deleted from osteogenic progenitors and circulating DKK1 levels are suppressed[42]. These results suggest that local, rather than circulating DKK1 protein levels dictate net biological outcomes. Moreover, cell-intrinsic (cell-autonomous) functions of DKK1 may exist. Consistent with this notion, membrane-bound forms, as well as nuclear abundance of DKK1, have been described in T-cells and human colorectal cancer biopsies, respectively[21,43] and supplementation of recombinant DKK1 failed to recover hematopoietic defects in mice with genetic DKK1 deficiency[2]. Likewise, the DKK1 protein was readily detectable in the nucleus of cancer cells in our study and DKK1-controlled inflammation was relayed in a cell-autonomous fashion, irrespective of soluble DKK1 abundance. Posttranscriptional (e.g., alternative splicing), as well as post-translational (e.g., glycosylation) events, may determine the fate of DKK1 within the cell and confer distinct biological functions. Additional studies are required to understand the biochemistry of DKK1 isoforms. It will be of particular interest to decipher whether the nuclear appearance of DKK1 is a cancer-specific phenomenon or also detectable in non-malignant cells. We currently do not know how and under which conditions DKK1 is retained within the cell. We also did not dissect how DKK1

mechanistically impinges on SOCS3, nor did we evaluate why DKK1 overexpression is insufficient to evoke an inflammatory response. Finally, the triggers underlying high DKK1 expression in cancer remain to be defined.

We observed a transient suppression of DKK1 serum levels in response to microbial inflammation in mice and humans, while DKK1 tissue levels were concurrently increased. These changes followed spatiotemporal patterns as evidenced by increased, unchanged and time-delayed LPS-induced DKK1 accumulation in the spleen, bone, and bone marrow, respectively. This pattern resembles platelet kinetics during infection, which are consumed in the circulation, degraded in the spleen and replenished by the bone marrow[44]. Consistent with this notion, changes in circulating platelet counts paralleled those in DKK1 serum concentration in patients and rodents. We recently reported similar associations in patients with Covid-19[45]. Although our data implicate contributions of p38-signaling to this process, further mechanistic studies are required to comprehensively characterize how DKK1 biology is reshaped during innate immune responses, particularly in platelets.

Collectively, our findings question whether circulating DKK1 concentration is fully reflective of DKK1 activity. This is particularly important in cancer, where DKK1 has been evaluated as a prognostic biomarker with ambiguous results[6,12,16,46,47]. Likewise, neutralizing soluble DKK1 may prove ineffective in holistically blocking DKK1 biology in the treatment of malignant and non-malignant diseases, which should be further explored experimentally.

Genes involved in immune responses are typically polymorphic and diverse, which allows for defending a broad range of pathogens in a highly specific manner[48]. Vice versa, immune diversity confers a risk of developing immunopathology and complicates the implementation of targeted therapies in the clinic. Our study identified multiple genetic DKK1 variants associated with increased cytokine production in humans and DKK1 serum levels were highly variable in our cohort of patients with pneumonia. As such, DKK1 expression may belong to a group of traits conferring a more effective immune response (benefit), while increasing the susceptibility to excessive inflammation and resultant organ damage (cost). Whether reduced inflammatory cytokine production upon DKK1 deficiency entails impaired disease resistance (elimination of pathogens) as observed in mice with tumor necrosis alpha knock-out[49] remains to be determined in experimental models involving living bacteria (e.g., cecal ligation and puncture).

Finally, our study adds DKK1 to a group of peptides, which link bone metabolism to innate immunity such as RANKL, OPG or FGF23[50–53]. Understanding the evolutionary origins and benefits of interweaving these two traits may help to create a broader perspective on bone biology in vertebrates. In view of our own results as well as those of others[2,35,54], we propose that DKK1 controls immunity via three distinct mechanisms: (1) direct and indirect modulation of inflammation, (2) regulation of leukopoiesis and 3.) anatomical shaping of the hematopoietic niche via effects on bone metabolism.

In summary, we show that physiological and pathological DKK1 activity promotes inflammatory cytokine responses in mice and humans. We suggest that exploiting this knowledge will prove useful in future studies.

## Methods

**Mouse models**. C57BL/6 J $Dkk1^{fl/fl}$ mice with loxP sites flanking exons 1 and 2 of the Dkk1 locus[55] were crossed with C57BL/6 J Rosa26-CreERT2 mice to obtain $Dkk1^{fl/fl}$:Rosa26-CreERT2 allowing for tamoxifen-inducible global deletion of Dkk1 ($Dkk1^{-/-}$). Cre-negative littermates were used as controls (referred to as wildtype; WT). Animals of both genotypes were injected with 100 µl of tamoxifen

(10 g/l) dissolved in sunflower oil for 5 consecutive days at the age of 5–8 weeks and used for in vivo experiments 2 weeks thereafter. Mice with osteoblast/osteocyte-specific Dkk1 knock-out (Dkk1$^{\Delta Dmp1}$) were generated by crossing Dkk1$^{fl/fl}$ mice with Dmp1:Cre transgenic animals, while Dkk1$^{fl/fl}$ mice were crossed with LysM:Cre transgenic animals to yield animals with myeloid cell-specific Dkk1 deletion. Cre-negative littermates served as controls for all experiments.

C57BL/6 J wild-type mice were purchased from Janvier Laboratories (Le Genest-Saint-Isle, France) and used at the age of 8–10 weeks. All mice were housed in groups of 3–6 at the animal facility of the Technical University of Dresden and kept on a 12 h light:dark cycle. Gram-negative sepsis was modeled by i.p. administration of sublethal doses of LPS (5 mg/kg) (Sigma-Aldrich, St.Louis, MO) to mice of the respective genotypes. Controls received an equivalent volume of water. Mice were sacrificed for blood and organ collection at the indicated time points. Circulating markers of organ damage markers were analyzed in serum samples by routine testing at the Institute of Clinical Laboratory Medicine of the Technical University of Dresden. To determine mortality outcomes, mice received a high dose of LPS (15 mg/kg) and were continuously monitored by trained staff (3x/day). Mice were sacrificed upon reaching human endpoints as defined by local authorities.

In vivo neutralization of circulating DKK1 was achieved by a single injection of the monoclonal anti-DKK1 antibody DKN01 kindly provided by Leap Therapeutics (Cambridge, MA) at 5 and 10 mg/kg, respectively, 18 h prior to LPS exposure as previously described[14,15,56]. For p38 inhibition, C57BL/6 J mice were treated with ralimetinib dimesylate (LY22288220) (Medchemexpress, South Brunswick, NJ) at 5 mg/kg 1 h prior to LPS injection. All animal experiments were approved by the Landesdirektion of Sachsen and performed according to local ethics regulations.

**Cell culture**. Human cancer cell lines (PC3, MDAMB231, MCF7, T47D, and SaoS2) were obtained from the German Collection of Microorganisms and Cell Cultures (DSMZ). Gastrointestinal cancer cell lines (LoVo, Caco-2, Huh-7, and HepG2) were kindly provided by Prof. Zeissig (TU Dresden), while DU-145 and LNCaP cells were a gift from Prof. Dubrovska (TU Dresden). Prostate cancer cell lines were grown in RPMI supplemented with 10% FCS and 1% penicillin/streptomycin, while breast cancer cell lines received DMEM-F12 (all from Gibco, Waltham MA). Gastrointestinal cancer cells were grown in DMEM or RPMI, as appropriate. The following reagents were used: DMSO, Lipopolysaccharides from E.coli O55:B5 (both from Sigma-Aldrich, St. Louis, MO), Leptomycin B, Bortezomib, Cycloheximide, XAV939 (all from Selleckchem, Houston, TX), high-molecular-weight Poly I:C (Invivogen, San Diego, CA), recombinant human Dickkopf1, Interleukin 1 beta, tumor necrosis factor-alpha (all from Peprotech, Rocky Hill, NJ), and recombinant Wnt3a (R&D, Minneapolis, MN). Recombinant human DKK1 was further purchased from R&D (Minneapolis, MN). DKK1-neutralizing antibodies and isotype controls were obtained from R&D (Minneapolis, MN) and Leap Therapeutics (Cambridge, MA), respectively. Cell cultures were maintained under a humidified atmosphere at 37 °C and 5% CO$_2$.

**RNA sequencing and bioinformatic analysis**. RNA sequencing was performed at the Genome Center of the Technical University of Dresden as previously described in ref.[57]. Three independent biological replicates per genotype (siCtrl and siDKK1, respectively) were analyzed. Low-quality nucleotides were removed by Illumina fastq filter (http://cancan.cshl.edu/labmembers/gordon/fastq_illumina_filter/). Reads were further subjected to adapter trimming using cutadapt[58]. Alignment of the reads to the human genome was performed using STAR Aligner[59] and the parameters: "--runMode alignReads --outSAMstrandField intronMotif --outSAMtype BAM SortedByCoordinate--readFilesCommand zcat". Human Genome version GRCh38 (release M12 GENCODE) was used for the alignment. The parameters: "htseq-count -f bam -s reverse -m union -a 20", HTSeq-0.6.1p1[60] were used to count the reads that map to the genes in the aligned sample files. The GTF file (gencode.v34.annotation.gtf) used for reads quantification was downloaded from Gencode (https://www.gencodegenes.org/human/release_34.html). Gene-centric differential expression analysis was performed using DESeq2_1.8.1[61]. Pathway and functional analyses were performed using GSEA[62] and EGSEA[63], respectively. GSEA is a stand-alone software with a graphic user interface (GUI). To run GSEA, a ranked list of all the genes from DESeq2-based calculations was created by taking the −log10 of the p value and multiplying it with the sign of the fold-change. This ranked list was then queried against Molecular Signatures Database (MSigDB), Reactome, KEGG, and GO-based repositories. EGSEA is a R/Bioconductor-based command-line package. For functional analyses using EGSEA, a gene-based raw count expression matrix was utilized. The same database repositories as above were used for performing the functional analyses. To delineate causal upstream regulatory pathways, that entail the observed changes in gene expression, we performed further analyses using CARNIVAL[23]. Filtered (genes whose expression values add up to more than 10 across the samples) and normalized gene expression matrix was used to calculate Transcription Factor (TF) and Pathway activity scores by using DoRothEA[64] and PROGENy[65], respectively. Next, "OmnipathR"[66] was used to construct a whole organism protein–protein interaction (PPI) network. All these data along with the list of differentially expressed genes were used as inputs for CARNIVAL to create a causal regulatory network.

**Human genetic analysis**. SNPs associated with DKK1 were downloaded from dbNSP[28]. Cytokine Quantitative Trait Loci (cQTLs) data were extracted from the previously published 500FG cohort study (https://hfgp.bbmri.nl/)[29]. These two datasets were intersected to determine cQTLs associated with DKK1. In an independent analysis, pQTLs (protein Quantitative Trait Loci) associated with DKK1 were extracted from available datasets (https://www.omicscience.org/apps/pgwas/)[30]. Specifically, genes that are targets of pQTLs associated with DKK1 were selected. Cytoscapev3.8.2 was used to render a bipartite network between the pQTLs and target genes extracted from reference [30]. The pQTLs and target genes (cytokines/chemokines) form the nodes of this network. The edges represent the probable effect of the QTLs on the target gene. Red-colored edges reflect positive effects (increase in expression) and blue-colored edges visualize negative effects (decrease in expression).

**Human pneumonia cohort**. After obtaining local institutional review board approval (Ref.EK 191052016), patients (age >18 and <90 years) admitted to the University Hospital Dresden with a clinical and/or radiographical diagnosis of pneumonia underwent repeated blood sampling. Patients who received antibiotics prior to admission were excluded from the study. In total, 27 patients were included in the study. Blood samples were obtained within the first 48 h of hospitalization (referred to as "initial") and on subsequent days thereafter (>day 3; referred to as "follow-up"). Maximum follow-up was 7 days. Each figure shows the last individually available measure of the corresponding variable. Vital signs and clinical parameters were documented by medical staff daily. Additional laboratory data such as C-reactive protein (CRP) and Procalcitonin (PCT) levels were measured if possible. Circulating DKK1 levels were determined using a commercially available ELISA kit (Biomedica, Vienna, Austria).

**Isolation and differentiation of murine osteoblasts and bone marrow-derived macrophages**. Primary murine osteoblasts and macrophages were differentiated from bone marrow-resident stem cells. Briefly, femora from 12- to 14-week-old mice were flushed with RPMI or DMEM followed by erythrocyte lysis using ACK lysis buffer (all from Gibco, Waltham, MA). Cells were counted and seeded at a density of $10 \times 10^6$ and $2 \times 10^6$/ml, respectively. Macrophages were obtained by exposing cells to a differentiation medium for 7 days consisting of RPMI supplemented with 10% FCS, 1% penicillin/streptomycin, 2 mM glutamine, non-essential amino acids, and 30% L929 conditioned medium as a source for M-CSF. Osteoblasts were differentiated by growing cells in DMEM supplemented with 10% FCS and 1% P/S for 7 days followed by a 7-day exposure to a differentiation medium consisting of DMEM supplemented with 10% FCS, 1% P/S, beta-glycerol-phosphate (5 mM), and ascorbic acid 2-phosphate (100 μM). The medium was changed twice weekly. After 7–10 days, differentiated murine osteoblasts were used for in vitro experiments.

**Protein isolation and immunoblotting**. Cells were lysed in RIPA buffer supplemented with protease and phosphatase inhibitors (Roche Applied Science, Mannheim, Germany). Cell membranes were disrupted by sonication in a water bath and supernatants were collected following centrifugation. Protein concentration was assessed by BCA assay (Pierce Technologies, Waltham, MA). Immunoblotting was performed according to standard protocols. Briefly, equal amounts of protein (8–20 μg) were denatured at 95° in Laemmli buffer, resolved on SDS page and separated by electrophoresis, followed by transfer onto a 0.2 μm nitrocellulose membrane. After blocking the membranes in 5% BSA, primary antibodies were applied and incubated overnight at 4°. Signals were visualized using HRP-conjugated secondary antibodies (R&D, Minneapolis, MN) and ECL western blotting substrate (Pierce Technologies, Waltham, MA). The following primary antibodies were used: anti-beta-Tubulin (1:1000, Cell Signaling Technology, #2146), anti-DKK1 (1:500, R&D, #AF1096), anti-Lamin A/C (1:1000, Santa Cruz Biotechnology, #sc-376248), anti-SOCS1 (1:500, Cell Signaling Technology, #3950), anti-SOCS3 (1:500, Santa Cruz Biotechnology, #sc-73045), anti-SOCS3 (1:500, Cell Signaling Technology, #52113), anti-RelA/p65 (1:1000, Cell Signaling Technology, #8242), anti-RelA/p65-phospho (1:1000, Cell Signaling Technology, #3033), anti-p38 (1:1000, Cell Signaling Technology, #9212), and anti-p38-phospho (1:1000, Cell Signaling Technology, #4511). Band intensities were quantified by ImageJ.

**Enzyme-linked sorbent assays (ELISAs)**. Protein levels of selected cytokines, chemokines and DKK1 in cell culture supernatants, mouse or human serum were quantified using commercially available enzyme-linked sorbent assays (ELISA) kits according to the manufacturer's instructions. The following kits were used: human DKK1 (Biomedica, BI-20413), mouse Dkk1 (Abcam, ab197746), mouse Interleukin 1 beta (R&D, DY401), human Interleukin 1 beta (R&D, DY201), human CXCL8 (R&D, D8000C), and mouse Tnfa (R&D, DY410).

Tissue levels of selected mediators were determined according to previously published protocols[67]. Protein was isolated from different tissues using T-PER supplemented with HALT protease and phosphatase inhibitor cocktail (Pierce Technologies, Waltham, MA) followed by homogenization using TissueLyzer (Qiagen, Hilden, Germany) and stainless beads. Corresponding total protein concentrations were determined by BCA assay (Pierce Technologies, Waltham, MA). Tissue lysates were diluted appropriately and protein levels of cytokines and DKK1 were quantified by ELISA. The abundance of the respective molecules was

normalized to the corresponding total protein levels and expressed as pg/mg of protein. The same protocol was used for cytokine measurements from cell lysates.

**Cytokine proteome profiler array.** Simultaneous measurement of multiple cytokines and chemokines in spleen lysates was achieved by Proteome Profiler Mouse Cytokine Array Kit, Panel A (R&D, Minneapolis, MN). The assay was performed according to the manufacturer's instructions. Nitrocellulose membranes precoated with cytokine/chemokine-specific capture antibodies were blocked, followed by incubation with tissue lysates and detection antibodies at 4° under steady shaking overnight. Membranes were washed, Streptavidin-HRP was added and signals were visualized using enhanced chemoluminescence. Signal intensities were quantified by ImageJ.

**Cytokine profiler PCR array.** Cytokine expression in PC3 cells was screened using the RT$^2$ profiler PCR array for pro-inflammatory cytokines and receptors (Qiagen, Hilden, Germany) according to the manufacturer's protocol. Results were calculated utilizing a software template provided by the supplier, whereby the relative expression of the target genes in DKK1-deficient cells was normalized to the respective expression in wild-type controls. GAPDH served as the housekeeping gene.

**RNA interference.** RNA interference-mediated knockdown of selected targets was achieved by transfecting cells with commercially available small interfering RNAs (siRNAs). The following oligonucleotides were used (all from Thermo Fisher Scientific, Waltham, MA): siDKK1 (s22721, referred to as "siRNA#1"), siDKK1 (s22723, referred to as "siRNA#2"), siLRP5 (s8294), siLRP6 (s8291), siTLR4 (s14194), siMyD88 (s9236), siSOCS3 (s17191), siSOCS1 (s16470), siNFKBIA (s530885), and siNFKBIB (s194652). Cells were washed with HBSS before being exposed to the respective siRNAs at a final concentration of 50 nM diluted in DharmaFECT (Horizon Discovery, Cambridge, GB) and OptiMEM GlutaMAX (Gibco, Thermo Fisher Scientific, Waltham, MA). Cells transfected with non-targeting (scrambled) siRNA served as controls. The culture medium was changed 6 h after siRNA transfection and cells were used for further experiments 24 or 48 h thereafter as indicated.

**Subcellular protein fractionation.** Protein extraction from different cellular compartments was achieved using a subcellular protein fractionation kit (Pierce Technologies, Waltham, MA) according to the manufacturer's instructions. Samples were used for conventional immunoblotting as described elsewhere in this section. Fractionation efficacy was assessed by analyzing the abundance of compartment-specific proteins (Lamin A/C, Tubulin).

**Immunofluorescence staining.** PC3 cells were seeded in 12-well plates and grown for 24 h before siRNA transfection was performed as described elsewhere in this section. Cells were washed with PBS followed by fixation in 4% paraformaldehyde. After a 5 min wash in PBS, cells were permeabilized using 0.5% Triton X-100 (Sigma-Aldrich, St. Louis, MO). Samples were blocked in 1% BSA for 1 h at room temperature to minimize unspecific binding followed by probing with primary antibodies, which were diluted 1:200 in 1% BSA and incubated at 4 °C overnight. Following several washing steps, fluorescence-labeled secondary antibodies were applied (1:1000) for 30 min at room temperature. Cells were washed again before being exposed to 4,6-Diamidin-2-phenylindol (DAPI) for 5 min to allow nuclear staining. Following another wash, signals were visualized using a confocal microscope (Zeiss LSM 880 AiryScan, Carl Zeiss Vision, Oberkochen, Germany).

**RNA extraction, cDNA synthesis, and qPCR analysis.** RNA extraction from cultured cells was performed using the RELIA Prep Kit (Promega Corp., Fitchburg, WI) and the High Pure RNA Isolation Kit (Roche Applied Sciences, Mannheim, Germany), according to the manufacturer's instructions. For RNA isolation from murine osteoblasts, TRIzol reagent (Thermo Fisher Scientific, Waltham, MA) was used. Complementary DNA (cDNA) was synthesized from 500 ng RNA by reverse transcription using random primers (Thermo Fisher Scientific, Waltham, MA), dNTPs (Carl Roth GmbH, Karlsruhe, Germany), MMLV RT, and RNasin (both from Promega Corp., Fitchburg, WI). Complementary DNA (cDNA) was diluted in nuclease-free water and used for subsequent qPCR analysis. Here, cDNA was mixed with GoTaq Mastermix (Promega Corp., Fitchburg, WI), 10 μM primers (forward and reverse, respectively) and ddH2O. Carboxy-Rhodamine Dye (CXR) (Promega Corp., Fitchburg, WI) served as the reference dye. All qPCR analysis were performed on a StepOnePlus$^{TM}$ cycler (Applied Biosystems, Carlsbad, CA). Each primer set was validated by melting curve analysis and samples were run in duplicates. Primer sequences are listed in Table S2. Relative mRNA expression of selected targets was calculated using the ΔΔCT method. *GAPDH* and *Actb* served as housekeeping genes for human and mouse samples, respectively.

**RNA sequencing data from primary human tumors.** Associations between transcript levels of DKK1 and pro-inflammatory cyto- and chemokines in primary human tumor tissues were investigated in a publicly available RNA sequencing dataset (The Cancer Genome Atlas) using the GEPIA web tool[19], whereby Spearman's correlation

was used to determine such correlations. Results were visualized using Prism (Graphpad Inc., LaJolla, CA). Likewise, the expression of DKK1, IL1B, and CXCL8 were determined in an independent cohort of primary prostate cancer samples using a commercially available prostate cancer cDNA array (Origene Technologies, Rockville, MD). Here, GAPDH served as the housekeeping control.

**Overexpression of DKK1 and RELA.** The full open reading frame (ORF) sequences of human and mouse DKK1, as well as RELA, were overexpressed in the indicated cell types using commercially available plasmids (Sino Biological, Beijing, China and Addgene, Watertown, MA) using FugeneHD transfection reagent (Promega Corp., Fitchburg, WI). Cells were exposed to 0.5 μg of plasmid diluted in FugeneHD and OptiMEM GlutaMAX (Gibco, Thermo Fisher Scientific, Waltham, MA) or the corresponding empty backbone vector for 5 h. Further experimental procedures were conducted on the following day.

**Ex vivo tissue culture.** Femora and humera were collected in PBS for 3 h following intraperitoneal LPS injection, flushed to remove the bone marrow and weighed, followed by 24 h cultivation in a culture medium. Supernatants were collected and subjected to cytokine measurements by ELISA. Results were normalized according to the respective tissue weight and expressed and pg/g of tissue.

**Statistical analysis and reproducibility.** Experimental data is expressed as mean ± SEM and comprises ≥3 biologically independent replicates if not stated otherwise. No sample size calculations were performed prior to conducting in vitro or in vivo experiments. Sample sizes were based on previously published literature and individual experience. Comparisons between groups of two were performed by unpaired, two-tailed student's *t*-test. Before-after comparisons in human subjects were performed by the Wilcoxon-signed-rank test. Groups of three or more were analyzed by ordinary one-way ANOVA followed by Holm–Sidak's post hoc test adjusted for multiple comparisons. Spearman's correlation analysis was used to investigate associations between transcripts in human tumor tissues, while linear regression analysis (least square fit) was applied to assess correlations between selected variables in human serum samples. Major experimental findings were confirmed by independent researchers and reproduced successfully. Statistical significance was assumed at *p* values <0.05. Grubb's Test (alpha = 0.05) allowed for the exclusion of one statistically significant outlier per dataset. Prism V9 (Graph-Pad Inc., LaJolla, CA) was used for data analysis and visualization.

**Reporting summary.** Further information on research design is available in the Nature Portfolio Reporting Summary linked to this article.

## Data availability

All source data are provided in the respective files (Supplementary Data 2 and 3). RNA sequencing results are available at Gene Expression Omnibus (GEO) via accession number GSE217231.

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

## Acknowledgements

We thank Dorit Breining, Vanessa Passin, Heidi Lunze, and the Bone Lab staff for their technical assistance. This work was supported by grants from the DFG priority program µBONE 2084 to T.D.R., A.G., M.R., L.C.H., B.W., as well as DFG FOR 5146 FerrOs to M.R. and L.C.H. Further support was provided by the DFG Heisenberg program to B.W. T.D.R., A.G., S.P., and N.P.J. received support from Mildred-Scheel-Nachwuchszentrum Dresden, while N.P.J. was further funded by the German Academic Scholarship Foundation. T.E.A. and H.T. were supported by grants from the Austrian Science Fund (FWF P33070) and the Austrian Research Promotion Agency FFG (COMET), respectively. AW received funding from the NIH (T32 AR07107-39 and K08AI128745). The authors thank Kristin Eismann (TP Molecular Analysis/Mass Spectrometry, CMCB, TU Dresden) for excellent technical assistance. The TP Molecular Analysis/Mass Spectrometry was supported by grants from the German Federal Ministry of Education and Research (BMBF) program "Unternehmen Region" (03Z2ES1 and 3Z22EB1), the German Research Foundation (INST 269/731-1 FUGG), and the European Regional Development Fund (ERDF/EFRE) (100232736).

## Author contributions

N.P.J. and T.D.R. conceptualized and designed the study and wrote the manuscript with assistance from A.W., T.E.A., A.G., J.S., B.W., H.T., M.R., and L.C.H. N.P.J. and S.P.

performed in vitro, in vivo experiments, and data analysis. S.T., M.L.C., and M.H. assisted with mouse studies and in vitro experiments. A.S. analyzed RNAseq data, genetic datasets, and performed bioinformatic analysis. A.K. collected clinical specimens and patient data.

## Funding

## Competing interests

M.R. is an editorial board member of Communications Biology, but was not involved in the editorial review of, nor the decision to publish this article. The remaining authors declare no competing interests.
