## [Peer Review File · Communications Biology]

Reviewers' comments:

Reviewer #1 (Remarks to the Author):

I feel sorry for the late review of the article firstly. The authors have conducted levels of experiments to prove DKK1 fuels inflammatory cytokine responses, including bioinformatics, cell model, pneumonia clinical samples, and animal endotoxemia. I would like to comment the rigorous experimental design and the additional experiments has greatly improved the quality of the manuscript.

Inflammatory cytokine responses are primarily defense response to Pathogen- and damage-associated molecular patterns in immune cells, and the authors reported that neutralization of soluble DKK1 using monoclonal antibodies had no effect in bone-marrow-derived macrophages. I would like to recommend knockdown or knockout of DKK1 in macrophages to prove the pro-inflammatory role of DKK1.

In addition, like the Reviewer 1 in the original edition, I would like recommend DKK1 signaling in multiple cancer cell lines to avoid cell line-based biases, especially in immune cell lines, such as THP-1 monocytes.

Moreover, since osteoblasts are not defense cells in nature and very few articles reported its formation of NLRP3 inflammasome and pyroptosis, and osteoblasts-specific DKK1 knockdown cannot prove the DKK1-driven cytokine response. The authors may address the drawback in the Discussion.

Reviewer #2 (Remarks to the Author):

The manuscript tries to address the overarching role of Dickkopf1 in inflammatory diseases. Despite large amount of data that are presented, the manuscript appears to be convoluted, unfocused, and thus hard to read and to understand.

Major concerns:

The authors try to convince readers that DKK1 is important for both cancer and endotoxemia. While this might be interesting, the authors do not present an overarching scheme regarding the commonality of endotoxemia and cancer in a publishable manner. In fact, this manuscript is difficult to understand and the authors need to put significant amount of efforts to delineate these two seemingly different diseases. Endotoxemia is an acute disease while cancer is chronic inflammatory disease.

Due to ambitious and convoluted approach, the presentation of data lacked focus, and failed to reach comprehensive and in-depth mechanistic studies. Moreover, the concerning part of Figure 1 and 2 is that now the cohort/TCGA study about DKK1 and its implications in cancer is not novel any more. There are myriads of references showing that the elevated levels of DKK1 fueling inflammation in cancer.

Throughout the manuscript, scientific rigor is missing. For example, the authors failed to show the deletion efficiency of DKK1 conditional KO mice they generated. This makes Figure 6 hard to interpret. How the authors know whether physiologic responses they obtained in Figure 6 are because of DKK1 while they do not know the deletion efficiency?

Also, the authors used DKN-01, and there is no confirmation that the Ab neutralization worked. For siDKK1, the authors do not show quantitative decrease of DKK1 protein or DKK1 mRNA. A mere western blot data for DKK1 siDKK1 in Figure 1E will not be sufficient to explain other data in Figure 4.

For Figure 1, the authors do not indicate where exactly DKK1 size is located. DKK1 is located usually 28-35 KDa, and the blot is at most fuzzy. If all the bands in Figure 1E are

DKK1 bands, then the authors need to show the same for Figure 1F, and the blot does not convince this reviewer.

Figure 1G, DKK1 IF experiment does not look convincing. DKK1 is a soluble factor and therefore found mostly in cytosol or ER. But the authors argue otherwise. Unless the authors found nuclear localization signal in DKK1, the finding needs to be reanalyzed with care. The authors need much better resolution of IF image and perhaps use confocal microscopy as well as fractionation study to say about DKK1's location in the cell if that is of importance.

Point-to-point response: COMMSBIO-22-2502-T

We thank the reviewers for carefully and critically evaluating our manuscript. The helpful comments and suggestions provided by the referees have aided in further improving the quality of our work. Please find below the comments of the reviewers in **black** and our corresponding responses in **blue**.

Reviewer #1 (Remarks to the Author):

1. I feel sorry for the late review of the article firstly. The authors have conducted levels of experiments to prove DKK1 fuels inflammatory cytokine responses, including bioinformatics, cell model, pneumonia clinical samples, and animal endotoxemia. I would like to comment the rigorous experimental design and the additional experiments has greatly improved the quality of the manuscript.

We appreciate the reviewer's acknowledgment of our efforts to further improve the quality of our manuscript.

2. Inflammatory cytokine responses are primarily defense response to Pathogen- and damage-associated molecular patterns in immune cells, and the authors reported that neutralization of soluble DKK1 using monoclonal antibodies had no effect in bone-marrow-derived macrophages. I would like to recommend knockdown or knockout of DKK1 in macrophages to prove the pro-inflammatory role of DKK1.

We thank the referee for this suggestion. As siRNA-based gene knockdown in macrophages is technically very challenging and frequently ineffective¹, we created mice harboring a myeloid cell-specific *Dkk1* deletion. We isolated mBMDM from these mice and confirmed successful suppression of *Dkk1* expression by qPCR (Reviewer Fig. 1 and Fig. S7 F of our revised manuscript). Upon being challenged with LPS, *Dkk1* knock-out mBMDM (referred to as *Dkk1*^{ΔLysM}) displayed reduced inflammatory cytokine expression (exemplified by *Il1b*) compared to littermate controls (Reviewer Figure 1). These results our now included in our revised manuscript.

Reviewer Figure 1 (Fig S7 F of our revised manuscript). *Dkk1* transcript levels in isolated mBMDM from mice with myeloid cell specific *Dkk1* deletion (referred to as *Dkk1*^{ΔLysM}) and wildtype littermate controls (WT) measured by qPCR (n=7/genotype). The right panel shows *Il1b* mRNA levels in mBMDM isolated from the same mice exposed to water or LPS (50 ng/ml) for 6h (n=3/genotype and group).

We also followed the complementary genetic approach and overexpressed murine *Dkk1* in mBMDM isolated from wildtype C57BL/6J mice. Successful *Dkk1* overexpression was confirmed by qPCR (Reviewer Figure 2, left panel). Consistent with our results in human cancer cells with low basal DKK1 levels, ectopic *Dkk1* overexpression did not elicit transcription of inflammatory cytokines (Fig. S2 C of our revised manuscript). These data further support our notion, that DKK1 is necessary, but not sufficient to drive inflammation in cells, irrespective of malignant or non-malignant origin.

Reviewer Figure 2 (Fig. S2 C of our revised manuscript). Ectopic overexpression of the full open reading frame of murine DKK1 in murine bone marrow-derived macrophages (mBMDM) (n=4/genotype). Controls were exposed to an empty backbone vector (pCMV3). Data is shown as mean ± s.e.m. **p<0.01, ****p<0.0001 according to unpaired, two-tailed student's t-test.

3. In addition, like the Reviewer 1 in the original edition, I would like recommend DKK1 signaling in multiple cancer cell lines to avoid cell line-based biases, especially in immune cell lines, such as THP-1 monocytes.

Thank you for this comment. During the first round of revisions, we already included two additional cell lines (MDA-MB-231, DU-145) as well as primary murine cells (murine osteoblasts; mOB) to corroborate our findings and avoid cell line-based biases. We have now added additional data from mBMDM with genetic *Dkk1* deficiency (*Dkk1*^{ΔLysM}; see previous response).

According to the suggestion of the reviewer, we also conducted experiments with THP1 cells. We found that DKK1 expression was negligible in THP1 cells (cycle threshold/*C_T* in qPCR analysis= 33-37; GAPDH *C_T*=17-19), irrespective of whether we differentiated them with phorbol-12-myristat-13-acetat (PMA) or not (Reviewer Figure 3 A). Of note, successful differentiation was confirmed by microscopy (Reviewer figure 3, right panel). We then used a lipofectamine-based protocol recommended for nucleotide transfection of THP1 cells (<https://www.thermofisher.com/de/de/home/references/protocols/cell-culture/transfection-protocol/transfecting-plasmid-dna-into-thp-1-cells-using->

lipofectamine-ltx-reagent.html) but failed to reliably quantify DKK1 knockdown efficacy due to the extremely low expression under steady state conditions (Reviewer Figure 3 B).

We conclude that THP1 cells are a suboptimal model for studying DKK1 biology. Therefore, we refrained from including these data into our revised manuscript.

Reviewer Figure 3. (A) DKK1 transcript levels in human THP1 cells treated with or without phorbol 12-myristate 13-acetate (PMA) (50 ng/ml for 48h). The right panel shows light microscopy of the corresponding cells (B) DKK1 mRNA expression measured by qPCR in THP1 cells transfected with siRNA directed against DKK1 (siDKK1) or non-targeting control (siCtrl), followed by PMA differentiation.

Instead, we performed SOCS3-knockdown experiments in DKK1-deficient MDA-MB-231 cells and found that this fully restored the inflammatory response upon TLR4 ligation (Reviewer Figure 4 and Fig. S4 H), resembling our results in PC3 cells. These data provide additional evidence for a SOCS3-dependent, pro-inflammatory function of DKK1.

Reviewer Figure 4 (Fig. S4 H of our revised manuscript). IL1B mRNA levels in DKK1-competent and -deficient MDA-MBA-231 cells with or without SOCS3 knockdown treated with LPS (1µg/ml) or water for 24h. Data is shown as mean ± s.e.m. ****p<0.0001 according to one-way ANOVA with Holm-Sidak's post hoc test.

4. Moreover, since osteoblasts are not defense cells in nature and very few articles reported its formation of NLRP3 inflammasome and pyroptosis, and osteoblasts-specific DKK1 knockdown cannot prove the DKK1-driven cytokine response. The authors may address the drawback in the Discussion.

We thank the referee for this comment. We chose to study osteoblasts because these cells are the main endogenous source for DKK1. We inferred that deleting DKK1 from non-malignant cells with the highest physiological DKK1 expression would allow for efficient investigation of DKK1 biology, akin to our experimental approach in cancer cells.

Of note, we did not limit our experimental approaches to osteoblasts but also included a mouse with global *Dkk1* deletion, macrophages with *Dkk1* deficiency as well as human genetic data. Moreover, osteoblasts produce significant amounts of *Il1b* in response to LPS as shown in our *Dmp1:cre* mouse model (Fig. 6 D, E). We suggest that these findings collectively provide ample evidence for a pro-inflammatory function of *Dkk1*.

Reviewer #2 (Remarks to the Author):

The manuscript tries to address the overarching role of Dickkopf1 in inflammatory diseases. Despite large amount of data that are presented, the manuscript appears to be convoluted, unfocused, and thus hard to read and to understand.

Major concerns:

1. The authors try to convince readers that DKK1 is important for both cancer and endotoxemia. While this might be interesting, the authors do not present an overarching scheme regarding the commonality of endotoxemia and cancer in a publishable manner. In fact, this manuscript is difficult to understand and the authors need to put significant

amount of efforts to delineate these two seemingly different diseases. Endotoxemia is an acute disease while cancer is chronic inflammatory disease.

We thank the reviewer for pointing out the inherent differences in disease trajectories between cancer and septic shock (endotoxemia/LPS model). We agree that cancer-associated inflammation is chronic, while LPS-triggered inflammation is acute. However, both entities share similar molecular effector mechanisms. For example, organ damage and mortality in endotoxemia models is driven by inflammatory cytokine signaling^{2,3}. These cytokines are induced in response to TLR4 ligation and resultant NFkB activation. Likewise, many cancers exhibit constitutively active NFkB-signaling, which triggers permanent transcription of inflammatory cytokines and confers apoptosis resistance^{4,5}. Further, macrophages engage JAK/STAT signaling in response to LPS and the same pathway is frequently activated in malignant cells^{6,7}. Our study identifies another, previously unknown, molecular effector mechanism of inflammation, which is shared between malignant and non-malignant cells. We believe that this is a strength, rather than a weakness of our study.

We have highlighted these aspects in our revised manuscript for further clarification, which reads as follows:

“[...] Our results thus far suggested that inhibition of DKK1 activity may be exploited to restrain overshooting cytokine production, a hallmark of sepsis. In contrast to the chronic inflammation observed in cancer, sepsis is characterized by an acute cytokine storm. Nonetheless, both entities share similar molecular effector mechanisms, including NFkB activation¹⁸. We thus asked whether the pro-inflammatory function of DKK1 is conserved between malignant and non-malignant cells, irrespective of the disease setting [...]”

2. Due to ambitious and convoluted approach, the presentation of data lacked focus, and failed to reach comprehensive and in-depth mechanistic studies. Moreover, the concerning part of Figure 1 and 2 is that now the cohort/TCGA study about DKK1 and its implications in cancer is not novel any more. There are myriads of references showing that the elevated levels of DKK1 fueling inflammation in cancer.

We thank the reviewer for highlighting previously published literature on DKK1 and inflammation in cancer. We agree that the concept that DKK1 tunes anti-tumor immunity is now well established. In fact, these experimental studies were the basis for the initiation of clinical trials evaluating the efficacy of anti-DKK1 antibodies together with checkpoint inhibitors in various cancer entities. We recently discussed these and other findings in a review article⁸ and also highlighted them in our introduction (lines 101-104).

Nonetheless, our findings are novel in several aspects. First, we are – to the best of our knowledge- the first ones to describe a biological function of DKK1, which is independent of its soluble forms. Second, we show that DKK1 controls TLR4-related inflammation and

consequently, drives inflammatory cytokine production, organ damage and mortality in a mouse model of endotoxemia, which has not been reported previously. Moreover, we provide evidence for a SOCS3-dependent mechanism underlying these functions. We are also not aware of any study that has shown comprehensive correlation analysis between DKK1 and inflammatory cytokine transcript levels from the TCGA dataset. Finally, no studies have linked genetic DKK1 variants to inflammatory cytokine production in humans as demonstrated in Figure 5 of our manuscript. Likewise, our finding that circulating DKK1 parallels markers of clinical recovery in a cohort of patients with bacterial pneumonia is novel. Of note, we have recently reported on the prognostic value of DKK1 in patients with Covid-19⁹, highlighting the translational relevance of our mechanistic studies presented. As such, we suggest that our study adds important new insights into DKK1 biology in inflammation and innate immunity, which will be of interest to the community.

While we agree that mechanistic studies can always be further refined, we suggest that the SOCS3-/RelA-dependent mechanism outlined in our manuscript- which is independent of soluble DKK1- provides sufficient mechanistic data for the scope of our study. Of note, we have now confirmed this mechanistic insight in an independent cell line (see response to reviewer 1).

3. Throughout the manuscript, scientific rigor is missing. For example, the authors failed to show the deletion efficiency of DKK1 conditional KO mice they generated. This makes Figure 6 hard to interpret. How the authors know whether physiologic responses they obtained in Figure 6 are because of DKK1 while they do not know the deletion efficiency?

We thank the referee for this comment. As shown in supplementary figure 7 of our manuscript (Reviewer Figure 5, left panel), we confirmed that mice with conditional *Dkk1* knockout display diminished circulating *Dkk1* levels

Reviewer Figure 5 (Figure S7 of revised manuscript). (G) Circulating *Dkk1* levels in mice with tamoxifen-inducible, global DKK1 deletion (*Dkk1*^{-/-}) and WT controls (n=4 and n=3) measured by ELISA (H) *Dkk1* protein abundance in bone lysates from *Dkk1*^{-/-} mice and wildtype littermate controls (n=8/genotype) measured by ELISA. Data is shown as mean ± s.e.m. **p<0.01, ****p<0.0001 according to unpaired, two-tailed student's t-test.

To further validate these results, we isolated protein from the femora of *Dkk1*^{-/-} animals and WT littermate controls and determined *Dkk1* levels by ELISA. We failed to detect *Dkk1* protein in *Dkk1*^{-/-} animals, suggesting that our genetic approach is effective in

suppressing *Dkk1* production (Fig. S7 H of our revised manuscript). Together, these data confirm that *Dkk1*^{-/-} mice display decreased systemic (circulation) and local (bone) *Dkk1* levels.

4. Also, the authors used DKN-01, and there is no confirmation that the Ab neutralization worked. For siDKK1, the authors do not show quantitative decrease of DKK1 protein or DKK1 mRNA. A mere western blot data for DKK1 siDKK1 in Figure 1E will not be sufficient to explain other data in Figure 4.

a.) we confirmed successful DKK1 neutralization by DKN01 by showing that treatment with the antibody reversed Wnt-antagonism elicited by rDKK1 in primary murine osteoblasts. These results are included in Fig. 3K of our manuscript.

Reviewer Figure 6 (Figure 3K from our revised manuscript). mRNA levels of the Wnt-target gene *Lef1* in primary murine osteoblasts following treatment with recombinant Wnt3a (200ng/ml) with or without recombinant DKK1 (250ng/ml) and DKK1-neutralizing antibodies (DKN01 and anti-DKK1, respectively, both at 10µg/ml) (n=3/condition) *p<0.05; **p<0.01, ***p<0.001, ****p<0.0001. Data is expressed as mean ± SEM. [one-way ANOVA with Holm-Sidak's post-hoc test]

b.) According to Leap Therapeutics (the developer and supplier of DKN01), successful *in vivo* neutralization of *Dkk1* by the antibody is indicated by elevated circulating *Dkk1* levels measured by a commercially available ELISA. This is presumably explained by a prolonged half-life of antibody-captured *Dkk1*¹⁰. Accordingly, we followed this experimental approach and indeed found elevated circulating *Dkk1* levels in DKN01-treated animals (Reviewer Figure 7), suggesting that neutralization was successful. These results are now included in Fig. S7 B of our revised manuscript.

Reviewer Figure 7: Circulating DKK1 levels in C57BL/6J wildtype mice treated with DKN01 or isotype control (IgG) (n=15 and n=16). **p<0.01, two tailed student's t-test.

c.) In Figure 1 D, we confirm successful knockdown of DKK1 at mRNA level by qPCR as well as protein level (secreted DKK1 levels in supernatants, measured by ELISA). Additionally, we demonstrate DKK1 knockdown by western blotting as shown in Fig 1E, as well as immunofluorescence (Fig. 1 G). We respectfully suggest that 4 complementary experimental approaches (qPCR, ELISA, western blot and immunofluorescence) demonstrating strongly reduced DKK1 expression are sufficient to validate siRNA efficacy.

We have included an additional graph visualizing the quantification of our western blot results by ImageJ (Reviewer Figure 8 and Fig. S1 D)

Reviewer figure 8: Quantification of DKK1 band intensities from western blots by ImageJ (n=4 independent biological replicates per genotype). **p<0.01, one-way ANOVA with Holm-Sidak's post-hoc test.

5. For Figure 1, the authors do not indicate where exactly DKK1 size is located. DKK1 is located usually 28-35 KDa, and the blot is at most fuzzy. If all the bands in Figure 1E are DKK1 bands, then the authors need to show the same for Figure 1F, and the blot does not convince this reviewer.

Thank you for this comment. As shown in our western blot in Figure 1 E, DKK1 bands in cancer cell lysates range from approximately 42- 24 kDa (i.e. all bands shown). These bands are DKK1-specific as DKK1 knockdown abolishes all of them (Fig. 1 E). This is consistent with previous reports¹¹ as well as information provided by the supplier. As shown in Reviewer Figure 9 (derived from the R&D website), DKK1 protein produces distinct patterns in immunoblots with a "fuzzy appearance" among different cancer cell lines, suggestive of DKK1 isoform variability among these cells. We agree with the reviewer that the biology of DKK1 isoforms requires further experimental investigations.

Reviewer Figure 9. Dkk1 protein bands from different human cancer cell lines visualized by immunoblotting (derived from R&D website, accessed on October 11, 2022).

We also repeated our subcellular fractionation immunoblot (Figure 1F). The membrane including the protein ladder is shown in Reviewer Figure 10. Note that the membrane had to be cut right above 40 kDa to allow for detection of the loading control (tubulin). Consistent with previous blots, DKK1-specific bands range from 42 to approximately 24 kDa. The most prominent staining is observed around 38 and 33 kDa, which corresponds to the immunoblot shown in our figure 1F. We have updated our Figure 1 F accordingly.

Reviewer figure 10. Immunoblot from subcellular fractions of PC3 cells with (siDKK1) or without (siCtrl) DKK1 knockdown. The upper panel shows DKK1, the lower panel shows the loading (tubulin) and fractionation control (Lamin A/C).

6. Figure 1G, DKK1 IF experiment does not look convincing. DKK1 is a soluble factor and therefore found mostly in cytosol or ER. But the authors argue otherwise. Unless the authors found nuclear localization signal in DKK1, the finding needs to be reanalyzed with care. The authors need much better resolution of IF image and perhaps use confocal microscopy as well as fractionation study to say about DKK1's location in the cell if that is of importance.

We thank the reviewer for this comment. We repeated our immunofluorescence staining followed by high resolution confocal microscopy (40x) of wildtype and DKK1-deficient PC3 cells. Consistent with our previous observation, DKK1 protein was detected perinuclearly, likely corresponding to the ER. However, we again observed nuclear DKK1 staining (see merged picture). This signal is specific as it is not present in DKK1-deficient cells. (Reviewer figure 11 and Figure 1G of our revised manuscript). Likewise, DKK1 protein is detectable in nuclear fractions of cancer cells but not in nuclear fractions of their DKK1-deficient counterparts (see Figure 1 F and Reviewer Figure 10). We obtained similar results when we ectopically overexpressed DKK1 in T47D cells (Fig. S1 E). We suggest that this is not an artefact because our subcellular fractionation efficacy is high as indicated by nucleus restricted Lamin A/C expression. Moreover, nuclear presence of DKK1 has previously been reported by other groups¹².

Reviewer figure 11. Confocal microscopy (40x) of immunofluorescence staining of DKK1 in DKK1-competent (siCtrl) and -deficient (siDKK1) PC3 cells. DKK1 is shown in magenta, while the nucleus of the cells is stained by DAPI in blue.

Collectively, our data strongly point towards cell-intrinsic functions of DKK1, which are conserved between malignant and non-malignant cells. We suggest that our revised manuscript provides multiple lines of evidence supporting this concept.

References:

1. Dong, S.X.M., *et al.* Transfection of hard-to-transfect primary human macrophages with. *RNA Biol* **17**, 755-764 (2020).
2. Weber, G.F., *et al.* Interleukin-3 amplifies acute inflammation and is a potential therapeutic target in sepsis. *Science* **347**, 1260-1265 (2015).
3. Echtenacher, B. & Männel, D.N. Requirement of TNF and TNF receptor type 2 for LPS-induced protection from lethal septic peritonitis. *J Endotoxin Res* **8**, 365-369 (2002).
4. Suh, J., *et al.* Mechanisms of constitutive NF-kappaB activation in human prostate cancer cells. *Prostate* **52**, 183-200 (2002).
5. Ben-Neriah, Y. & Karin, M. Inflammation meets cancer, with NF-κB as the matchmaker. *Nat Immunol* **12**, 715-723 (2011).

6. Duncan, S.A., Baganizi, D.R., Sahu, R., Singh, S.R. & Dennis, V.A. SOCS Proteins as Regulators of Inflammatory Responses Induced by Bacterial Infections: A Review. *Front Microbiol* **8**, 2431 (2017).
7. Stoiber, D., *et al.* Lipopolysaccharide induces in macrophages the synthesis of the suppressor of cytokine signaling 3 and suppresses signal transduction in response to the activating factor IFN-gamma. *J Immunol* **163**, 2640-2647 (1999).
8. Jaschke, N., Hofbauer, L.C., Göbel, A. & Rachner, T.D. Evolving functions of Dickkopf-1 in cancer and immunity. *Cancer Lett* (2020).
9. Jaschke, N.P., *et al.* Circulating Dickkopf1 parallels metabolic adaptations and predicts disease trajectories in patients with Covid-19. *J Clin Endocrinol Metab* (2022).
10. Haas, M.S., *et al.* mDKN-01, a Novel Anti-DKK1 mAb, Enhances Innate Immune Responses in the Tumor Microenvironment. *Mol Cancer Res* **19**, 717-725 (2021).
11. Fleury, D., *et al.* Expression, purification and characterization of murine Dkk1 protein. *Protein Expr Purif* **60**, 74-81 (2008).
12. Aguilera, Ó., *et al.* Nuclear DICKKOPF-1 as a biomarker of chemoresistance and poor clinical outcome in colorectal cancer. *Oncotarget* **6**, 5903-5917 (2015).

REVIEWERS' COMMENTS:

Reviewer #1 (Remarks to the Author):

All the questions have been addressed, and the quality of the article has been greatly improved, and it's ready to be published.

Reviewer #2 (Remarks to the Author):

The authors' response to major concern 1 is misleading and therefore does not convince this reviewer at all. The authors presented superficial analyses of endotoxemia and cancer respectively, and thus failed to show what the manuscript aims to explain.

One of the key issues among many, is that the authors argue that they are the first ones to discuss biological functions of DKK1. The current manuscript does not clearly demonstrate how these two different forms of DKK1 can perform their biological functions distinctively with clarity.

This reviewer is not convinced that whether they meant nuclear DKK1 in cancer or endotoxemia, or both. From the most generous perspective, the overall conclusion that DKK1 support inflammation does not change, and thus it lacks thorough detailed mechanistic studies to delineate the difference between two different forms of DKK1.

The manuscript is misleading in that the expression of DKK1 in the TCGA database may be all from nuclear DKK1, and this reviewer does not see any biochemical/cell biological mechanistic analyses in depth to explain biological/functional differences between those two forms.

The authors failed to explore and demonstrate context-dependent role of DKK1. The authors argue their claims with cell line studies, but these are far from contemporary understanding of DKK1. Often, DKK1 is an immuno-suppressive ligand that suppresses NK cell function [Cell, 2016, 165:45]. As backed by several references, the role of DKK1 is proinflammatory, yet the direction or outcome of inflammation is completely different within cancer.

Also, the authors failed to explore important reference regarding the role of DKK1 in LPS-mediated inflammation [Blood, 2015, 126:2220] in that essentially the pro inflammatory role of DKK1 in LPS challenge in mice has been demonstrated. Furthermore, the use of Dmp1Cre Tg mice is not well explored and therefore lacking rigor.

The use of Tamoxifen complicates the immune responses since it affects immune functions [Curr Med Chem, 2009, 16:3076]. The authors did not explore/demonstrate any of these points and thus the manuscript remains superficial.